# Impact of Dataset Properties on Membership Inference Vulnerability of Deep Transfer Learning

## Abstract

We analyse the relationship between privacy vulnerability and dataset properties, such as examples per class and number of classes, when applying two state-of-the-art membership inference attacks (MIAs) to fine-tuned neural networks. We derive per-example MIA vulnerability in terms of score distributions and statistics computed from shadow models. We introduce a simplified model of membership inference and prove that in this model, the logarithm of the difference of true and false positive rates depends linearly on the logarithm of the number of examples per class. We complement the theoretical analysis with empirical analysis by systematically testing the practical privacy vulnerability of fine-tuning large image classification models and obtain the previously derived power law dependence between the number of examples per class in the data and the MIA vulnerability, as measured by true positive rate of the attack at a low false positive rate. Finally, we fit a parametric model of the previously derived form to predict true positive rate based on dataset properties and observe good fit for MIA vulnerability on unseen fine-tuning scenarios.

## 1 Introduction

Machine learning models are prone to memorising their training data, which makes them vulnerable to privacy attacks such as membership inference attacks (MIAs; Shokri et al., 2017; Carlini et al., 2022) and reconstruction attacks (e.g. Balle et al., 2022; Nasr et al., 2023). Differential privacy (DP; Dwork et al., 2006) provides protection against these attacks, but strong formal protection often comes at the cost of significant loss of model utility.

Finding the correct balance between making models resistant to attacks while maintaining a high utility is important for many applications. In health, for example, many European countries and soon also the EU within the European Health Data Space have requirements that models trained on health data that are made publicly available must be anonymous, i.e. they must not contain information that can be linked to an identifiable individual. On the other hand, loss of utility of the model due to privacy constraints may compromise the health benefits that might be gained from it.

In this paper, our aim is to theoretically understand and systematically apply two state-of-the-art MIAs, LiRA (Carlini et al., 2022) and RMIA (Zarifzadeh et al., 2024), to help understand practical privacy risks when fine-tuning deep-learning-based classifiers without DP protections. We focus on transfer learning using fine-tuning because this is increasingly used for all practical applications of deep learning and especially important when labeled examples are limited, which would often be the case in privacy-sensitive applications. Our case study focuses on understanding and quantifying factors that influence the vulnerability of non-DP deep transfer learning models to MIA. In particular, we theoretically study the relationship between the number of examples per class, which we denote as shots ($S$), and MIA vulnerability (true positive rate TPR at fixed false positive rate FPR) for a simplified model of fine-tuning and derive a power-law relationship in the form

$$\log(\text{TPR} - \text{FPR}) = -\beta_S \log(S) - \beta_0. \tag{1}$$

We complement the theoretical analysis with extensive experiments over many datasets with varying sizes in the transfer learning setting for image classification tasks and observe the same power-

law. This power-law has a practically remarkable implication that a practitioner could estimate the membership privacy risk and how large a training set would be needed to mitigate the risk.

**Related work** There has been evidence that classification models with more classes are more vulnerable to MIA (Shokri et al., 2017), models trained on fewer samples can be more vulnerable (Chen et al., 2020; Németh et al., 2023), and classes with less examples tend to be more vulnerable (Chang & Shokri, 2021; Kulynych et al., 2022; Tonni et al., 2020). Larger generalisation error, which is related to dataset size, has also been shown to be sufficient for MIA success (Song & Mittal, 2021), though not necessary (Yeom et al., 2018). Similarly, minority subgroups tend to be more affected by DP (Suriyakumar et al.; Bagdasaryan et al., 2019). Feldman & Zhang (2020) showed that neural networks trained from scratch can memorise a large fraction of their training data, while the memorisation is greatly reduced for fine-tuning. Additionally, Tobaben et al. (2023) reported how the MIA vulnerability of few-shot image classification is affected by the number of shots. Yu et al. (2023) studied the relationship between the MIA vulnerability and individual privacy parameters for different classes. Nonetheless, the prior works do not consider the rate of change in the vulnerability evaluated at a low FPR, as dataset properties change. Our work significantly expands on these works by explicitly identifying a quantitative relationship between dataset properties and MIA vulnerability (i.e., the power-law in Equation (1)).

**List of contributions** We analyze the MIA vulnerability of deep transfer learning using two state-of-the-art score-based MIAs, LiRA (Carlini et al., 2022) and RMIA (Zarifzadeh et al., 2024), which are a strong realistic threat model. We first analytically derive the power-law relationship in Equation (1) for both MIAs by introducing a simplified model of the optimal membership inference (Section 3). We support our theoretical findings by an extensive empirical study on the MIA vulnerability of deep learning models by focusing on a transfer-learning setting for image classification task, where a large pre-trained neural network is fine-tuned on a sensitive dataset.

1. *Closed-form per-example vulnerability:* We derive closed-form per-example LiRA and RMIA vulnerability (TPR at fixed FPR) in terms of MIA attack score distributions and statistics computed from shadow models (see Section 3.3).

2. *Power-law in simplified model of the optimal MI:* We formulate a simplified model of membership inference to quantitatively relate dataset properties and MIA vulnerability, in which LiRA is the optimal attack. For this model, we prove a power-law relationship between the per-example LiRA and RMIA vulnerability and the number of examples per class. We then extend the the power-law relationship to the average-case LiRA and RMIA vulnerability (See Section 3.4).

3. *Few-shot MIA:* We conduct a comprehensive study of MIA vulnerability (TPR at fixed low FPR) in the transfer learning setting for image classification tasks with target models trained using many different datasets with varying sizes and confirm the theoretical power law between the number of examples per class and the vulnerability to MIA (see Figure 1).

4. *Regression model:* We utilise our empirical observations to fit a regression model to predict MIA vulnerability ($\log(\text{TPR} - \text{FPR})$ at fixed low FPR) based on examples per class ($\log S$) and number of classes ($\log C$), which follows the functional form of the theoretically derived power-law. We show both very good fit on the training data as well as good prediction quality on unseen data from a different feature extractor and when fine-tuning other parameterisations (see Figure 4).

## 2 BACKGROUND

**Notation** for the properties of the training dataset $\mathcal{D}$: (i) $C$ for the number of classes (ii) $S$ for shots (examples per class) (iii) $|\mathcal{D}|$ for training dataset size ($|\mathcal{D}| = CS$). We denote the number of MIA shadow models with $M$.

**Membership inference attacks (MIAs)** aim to infer whether a particular sample was part of the training set of the targeted model (Shokri et al., 2017). Thus, they can be used to determine lower bounds on the privacy leakage of models to complement the theoretical upper bounds obtained through differential privacy.

**Likelihood Ratio attack** (**LiRA**; Carlini et al., 2022) While many different MIAs have been proposed (Hu et al., 2022), in this work we consider the Likelihood Ratio Attack (LiRA). LiRA is a strong attack that assumes an attacker that has black-box access to the attacked model, knows the

training data distribution, the training set size, the model architecture, hyperparameters and training algorithm. Based on this information, the attacker can train so-called shadow models (Shokri et al., 2017) which imitate the model under attack but for which the attacker knows the training dataset.

LiRA exploits the observation that the loss function value used to train a model is often lower for examples that were part of the training set compared to those that were not. For a target sample $(x, y_x)$, LiRA trains the shadow models: (i) with $(x, y_x)$ as a part of the training set ($(x, y_x) \in \mathcal{D}$) and (ii) without $x$ in the training set ($(x, y_x) \notin \mathcal{D}$). After training the shadow models, $(x, y_x)$ is passed through the shadow models, and based on the losses (or predictions) two Gaussian distributions are formed: one for the losses of $(x, y_x) \in \mathcal{D}$ shadow models, and one for the $(x, y_x) \notin \mathcal{D}$. Finally, the attacker computes the loss for the point $x$ using the model under attack and determines using a likelihood ratio test on the distributions built from the shadow models whether it is more likely that $(x, y_x) \in \mathcal{D}$ or $(x, y_x) \notin \mathcal{D}$. We use an optimization by Carlini et al. (2022) for performing LiRA for multiple models and points without training a computationally infeasible number of shadow models. It relies on sampling the shadow datasets in a way that each sample is in expectation half of the time included in the training dataset of a shadow model and half of the time not. At attack time each model will be attacked once using all other models as shadow models.

**Robust Membership Inference Attack** (**RMIA**; Zarifzadeh et al., 2024) Recently Zarifzadeh et al. (2024) proposed a new MIA algorithm called RMIA, which aims to improve performance when the number of shadow models is limited. Similar to LiRA, RMIA is based on shadow model training and computing the attack statistics based on a likelihood ratio. The main difference to LiRA is that RMIA does not compute the likelihood ratio based on aggregated IN/OUT statistics, but instead compares the target data point against random samples $(z, y_z)$ from the target data distribution. After computing the likelihood ratios over multiple $(z, y_z)$ values, the MIA score is estimated as a proportion of the ratios exceeding a preset bound. This approach makes RMIA a more effective attack when the number of shadow models is low.

**Measuring MIA vulnerability** Using the chosen MIA score of our attack, we can build a binary classifier to predict whether a sample belongs to the training data or not. The accuracy profile of such classifier can be used to measure the success of the MIA. More specifically, throughout the rest of the paper, we will use the true positive rate (TPR) at a specific false positive rate (FPR) as a measure for the vulnerability. Identifying even a small number of examples with high confidence is considered harmful (Carlini et al., 2022) and thus we focus on the regions of small FPR.

## 3 THEORETICAL ANALYSIS

In this section, we seek to theoretically understand the impact of the dataset properties on the MIA vulnerability. It is known that different data points exhibit different levels of MIA vulnerability depending on the underlying distribution (e.g. Aerni et al., 2024; Leemann et al., 2024). Therefore, we start with analysing *per-example* vulnerabilities for LiRA and RMIA. In order to quantitatively relate dataset properties to these vulnerabilities, a simplified model is formulated. Within this model, we prove a power-law between the per-example vulnerability and the number $S$ of examples per class. Finally, the per-example power-law is analytically extended to *average-case* MIA vulnerability, for which we provide empirical evidence in Section 4.

### 3.1 PRELIMINARIES

First, let us restate the MIA score from LiRA as defined by Carlini et al. (2022). Denoting the logit of a target model $\mathcal{M}$ applied on a target data point $(x, y_x)$ as $\ell(\mathcal{M}(x), y_x)$, the LiRA computes the MIA score as the likelihood ratio

$$\mathrm{LR}(x) = \frac{p(\ell(\mathcal{M}(x), y_x) \mid \mathbb{Q}_{\mathrm{in}}(x, y_x))}{p(\ell(\mathcal{M}(x), y_x) \mid \mathbb{Q}_{\mathrm{out}}(x, y_x))}, \tag{2}$$

where the $\mathbb{Q}_{\mathrm{in/out}}$ denote the hypotheses that $(x, y_x)$ was or was not in the training set of $\mathcal{M}$. Carlini et al. (2022) approximate the IN/OUT hypotheses as normal distributions. Denoting $t_x = \ell(\mathcal{M}(x), y_x)$, the score becomes

$$\mathrm{LR}(x) = \frac{\mathcal{N}(t_x; \hat{\mu}_{\mathrm{in}}(x), \hat{\sigma}_{\mathrm{in}}^2(x))}{\mathcal{N}(t_x; \hat{\mu}_{\mathrm{out}}(x), \hat{\sigma}_{\mathrm{out}}^2(x))}, \tag{3}$$

where the $\hat{\mu}_{\text{in/out}}(x)$ and $\hat{\sigma}_{\text{in/out}}(x)$ are the means and standard deviations for the IN/OUT shadow model losses for $(x, y_x)$. Larger values of $\text{LR}(x)$ suggest that $(x, y_x)$ is more likely in the training set and vice versa. Now, to build a classifier from this score, the LiRA tests if $\text{LR}(x) > \beta$ for some threshold $\beta$.

Next, let us restate how RMIA (Zarifzadeh et al., 2024) builds the MIA score. RMIA augments the likelihood-ratio with a sample from the target data distribution to calibrate how likely you would obtain the target model if $(x, y_x)$ is replaced with another sample $(z, y_z)$. Denoting the target model parameters with $\theta$, RMIA computes

$$\text{LR}(x, z) = \frac{p(\theta \mid x, y_x)}{p(\theta \mid z, y_z)}, \tag{4}$$

and the corresponding MIA score is given as

$$\text{Score}_{\text{RMIA}}(x) = \Pr_{(z, y_z) \sim \mathbb{D}} (\text{LR}(x, z) > \gamma), \tag{5}$$

where $\mathbb{D}$ denotes the training data distribution. Similar to LiRA, the classifier is built by checking if $\text{Score}_{\text{RMIA}}(x) > \beta$. In the following, we will use the direct computation of likelihood-ratio as described in Equation 11 of Zarifzadeh et al. (2024) which approximates $\text{LR}(x, z)$ using normal distributions.

## 3.2 COMPUTING THE TPR FOR LiRA AND RMIA

Using the LiRA formulation of Equation (3), the TPR for the target point $(x, y_x)$ for LiRA is defined as

$$\text{TPR}_{\text{LiRA}}(x) = \Pr_{\mathcal{D}_{\text{target}} \sim \mathbb{D}^{|\mathcal{D}|}, \phi^M} \left( \frac{\mathcal{N}(t_x; \hat{\mu}_{\text{in}}(x), \hat{\sigma}_{\text{in}}(x)^2)}{\mathcal{N}(t_x; \hat{\mu}_{\text{out}}(x), \hat{\sigma}_{\text{out}}(x)^2)} \geq \beta \mid (x, y_x) \in \mathcal{D}_{\text{target}} \right), \tag{6}$$

where $\beta$ is a threshold that defines a rejection region of the likelihood ratio test, $\hat{\mu}_{\text{in}}(x), \hat{\mu}_{\text{out}}(x), \hat{\sigma}_{\text{in}}(x)$ and $\hat{\sigma}_{\text{out}}(x)$ are LiRA statistics computed from shadow models, and $\phi^M$ denotes the randomness in shadow set sampling and shadow model training (see Appendix A for derivation).

For theoretical analysis of RMIA, we focus on the direct approach that is an approximation of the efficient Bayesian approach, as Zarifzadeh et al. (2024) empirically demonstrates that these approaches exhibit similar performances. Let $\hat{\mu}_{a,b}$ and $\hat{\sigma}_{a,b}$ denote, respectively, the mean and standard deviation of $t_b$ estimated from shadow models, where $a$ denotes which of $(x, y_x)$ and $(z, y_z)$ is in the training set. By Equation 11 in (Zarifzadeh et al., 2024), the per-example performance for RMIA is given as

$$\text{TPR}_{\text{RMIA}}(x) =$$

$$\Pr_{\mathcal{D}_{\text{target}} \sim \mathbb{D}^{|\mathcal{D}|}, \phi^M} \left( \Pr_{(z, y_z) \sim \mathbb{D}} (\text{LR}(x, z) \geq \gamma) \geq \beta \mid (x, y_x) \in \mathcal{D}_{\text{target}} \wedge (z, y_z) \notin \mathcal{D}_{\text{target}} \right) \tag{7}$$

$$\text{LR}(x, z) = \frac{\mathcal{N}(t_x; \hat{\mu}_{x,x}, \hat{\sigma}_{x,x}^2) \mathcal{N}(t_z; \hat{\mu}_{x,z}, \hat{\sigma}_{x,z}^2)}{\mathcal{N}(t_x; \hat{\mu}_{z,x}, \hat{\sigma}_{z,x}^2) \mathcal{N}(t_z; \hat{\mu}_{z,z}, \hat{\sigma}_{z,z}^2)}, \tag{8}$$

where $t_z$ is the score on $z$ similar to $t_x$ and $\phi^M$ denotes the randomness in shadow set sampling and shadow model training (see Appendix A for derivation).

We define the average-case TPRs for LiRA and RMIA by taking the expectation over the data distribution:

$$\overline{\text{TPR}}_{\text{LiRA}} = \mathbb{E}_{(x, y_x) \sim \mathbb{D}}[\text{TPR}_{\text{LiRA}}(x)] \tag{9}$$

$$\overline{\text{TPR}}_{\text{RMIA}} = \mathbb{E}_{(x, y_x) \sim \mathbb{D}}[\text{TPR}_{\text{LiRA}}(x)] \tag{10}$$

## 3.3 PER-EXAMPLE MIA VULNERABILITY

Although LiRA models $t_x$ by a normal distribution, we consider a more general case where the true distribution of $t_x$ is of the location-scale family. That is,

$$t_x = \begin{cases} \mu_{\text{in}}(x) + \sigma_{\text{in}}(x)t & \text{if } (x, y_x) \in \mathcal{D}_{\text{target}} \\ \mu_{\text{out}}(x) + \sigma_{\text{out}}(x)t & \text{if } (x, y_x) \notin \mathcal{D}_{\text{target}}, \end{cases} \tag{11}$$

where $t$ has the standard location and unit scale, and $\mu_{\text{in}}(x), \mu_{\text{out}}(x)$ and $\sigma_{\text{in}}(x), \sigma_{\text{out}}(x)$ are the locations and scales of IN/OUT distributions of $t_x$. We assume that the target and shadow datasets have a sufficient number of examples. This allows us to also assume that $\hat{\sigma}(x) = \hat{\sigma}_{\text{in}}(x) = \hat{\sigma}_{\text{out}}(x)$ and $\sigma(x) = \sigma_{\text{in}}(x) = \sigma_{\text{out}}(x)$, where $\hat{\sigma}(x)$ is the standard deviation of $t_x$ estimated from shadow models and $\sigma(x)$ is the true scale parameter of $t_x$. (See Appendix B for the validity of these assumptions). The following result reduces the LiRA vulnerability to the location and scale parameters of $t_x$.

**Lemma 1** (Per-example LiRA vulnerability). *Suppose that the true distribution of $t_x$ is of location-scale family with locations $\mu_{\text{in}}(x), \mu_{\text{out}}(x)$ and scale $\sigma(x)$, and that LiRA models $t_x$ by $\mathcal{N}(\hat{\mu}_{\text{in}}(x), \hat{\sigma}(x))$ and $\mathcal{N}(\hat{\mu}_{\text{out}}(x), \hat{\sigma}(x))$. Assume that an attacker has access to the underlying distribution $\mathbb{D}$. Then for a large enough number of examples per class and infinitely many shadow models, the LiRA vulnerability of a fixed target example is*

$$
\text{TPR}_{\text{LiRA}}(x) = \begin{cases} 1 - F_t\left(F_t^{-1}(1 - \text{FPR}_{\text{LiRA}}(x)) - \frac{\mu_{\text{in}}(x) - \mu_{\text{out}}(x)}{\sigma(x)}\right) & \text{if } \hat{\mu}_{\text{in}}(x) > \hat{\mu}_{\text{out}}(x) \\ F_t\left(F_t^{-1}(\text{FPR}_{\text{LiRA}}(x)) - \frac{\mu_{\text{in}}(x) - \mu_{\text{out}}(x)}{\sigma(x)}\right) & \text{if } \hat{\mu}_{\text{in}}(x) < \hat{\mu}_{\text{out}}(x), \end{cases} \tag{12}
$$

*where $F_t$ is the cdf of $t$ with the standard location and unit scale.*

*Proof.* See Appendix C.1. $\qquad\qquad\qquad\qquad\qquad\qquad\qquad\qquad\qquad\qquad\qquad\qquad\square$

Here we assume that an attacker trains shadow models with the true underlying distribution. However, in real-world settings the precise underlying distribution may not be available for an attacker. We relax this assumption in Appendix B so that the attacker only needs an approximated underlying distribution for the optimal LiRA as in Lemma 1.

Next we focus on the per-example RMIA performance. As in the case of LiRA, we assume that $t_x$ and $t_z$ follow distributions of the location-scale family. We have

$$
t_x = \begin{cases} \mu_{x,x} + \sigma_{x,x}t & \text{if } (x, y_x) \in \mathcal{D}_{\text{target}} \land (z, y_z) \notin \mathcal{D}_{\text{target}} \\ \mu_{z,x} + \sigma_{z,x}t & \text{if } (x, y_x) \notin \mathcal{D}_{\text{target}} \land (z, y_z) \in \mathcal{D}_{\text{target}} \end{cases} \tag{13}
$$

$$
t_z = \begin{cases} \mu_{x,z} + \sigma_{x,z}t & \text{if } (x, y_x) \in \mathcal{D}_{\text{target}} \land (z, y_z) \notin \mathcal{D}_{\text{target}} \\ \mu_{z,z} + \sigma_{z,z}t & \text{if } (x, y_x) \notin \mathcal{D}_{\text{target}} \land (z, y_z) \in \mathcal{D}_{\text{target}}. \end{cases} \tag{14}
$$

It is important to note that $\mu_{a,b}$ and $\sigma_{a,b}$ denote, respectively, a location and a scale, while previously defined $\hat{\mu}_{a,b}$ and $\hat{\sigma}_{a,b}$ are, respectively, a mean and a standard deviation. As for the analysis of LiRA, we assume that the target and shadow sets have a sufficient number of examples per class, and that $\sigma_x = \sigma_{x,x} = \sigma_{z,x}$, $\sigma_z = \sigma_{x,z} = \sigma_{z,z}$, $\hat{\sigma}_x = \hat{\sigma}_{x,x} = \hat{\sigma}_{z,x}$ and $\hat{\sigma}_z = \hat{\sigma}_{x,z} = \hat{\sigma}_{z,z}$, where $\sigma_x$ and $\sigma_z$ are, respectively, the true scales of $t_x$ and $t_z$, and $\hat{\sigma}_x$ and $\hat{\sigma}_z$ are, respectively, standard deviations of $t_x$ and $t_z$ estimated from shadow models (see Appendix B for the validity of these assumptions).

**Lemma 2** (Per-example RMIA vulnerability). *Suppose that the true distributions of $t_x$ and $t_z$ are of location-scale family with locations $\mu_{x,x}, \mu_{z,x}, \mu_{x,z}, \mu_{z,z}$ and scales $\sigma_x, \sigma_z$, and that RMIA models $t_x$ and $t_z$ by normal distributions with parameters computed from shadow models. For a large enough number of examples per class and infinitely many shadow models, the RMIA vulnerability of a fixed target example is bounded by*

$$
\text{TPR}_{\text{RMIA}}(x) \leq \begin{cases} 1 - F_t\left(F_t^{-1}(1 - \alpha) - \frac{\mathbb{E}_{(z,y_z)\sim\mathbb{D}}[q]}{\mathbb{E}_{(z,y_z)\sim\mathbb{D}}[A]}\right) & \text{if } \mathbb{E}_{(z,y_z)\sim\mathbb{D}}[A] > 0 \\ F_t\left(F_t^{-1}(\alpha) - \frac{\mathbb{E}_{(z,y_z)\sim\mathbb{D}}[q]}{\mathbb{E}_{(z,y_z)\sim\mathbb{D}}[A]}\right) & \text{if } \mathbb{E}_{(z,y_z)\sim\mathbb{D}}[A] < 0, \end{cases} \tag{15}
$$

*for some constant $\alpha \geq \text{FPR}_{\text{RMIA}}(x)$, where*

$$
q = \frac{(\mu_{x,x} - \mu_{z,x})(\hat{\mu}_{x,x} - \hat{\mu}_{z,x})}{\hat{\sigma}_x^2} - \frac{(\mu_{x,z} - \mu_{z,z})(\hat{\mu}_{x,z} - \hat{\mu}_{z,z})}{\hat{\sigma}_z^2} \tag{16}
$$

$$
A = \frac{\sigma_x}{\hat{\sigma}_x^2}(\hat{\mu}_{x,x} - \hat{\mu}_{z,x}) + \frac{\sigma_z}{\hat{\sigma}_z^2}(\hat{\mu}_{x,z} - \hat{\mu}_{z,z}). \tag{17}
$$

*Proof.* See Appendix C.2. $\qquad\qquad\qquad\qquad\qquad\qquad\qquad\qquad\qquad\qquad\qquad\qquad\square$

Note that here we must assume that the attacker has access to the underlying distribution for the optimal RMIA as the Equations (16) and (17) depend on the parameters computed from shadow models.

## 3.4 A SIMPLIFIED MODEL OF THE OPTIMAL MEMBERSHIP INFERENCE

Now we construct a simplified model of membership inference that streamlines the data generation and shadow model training.

We sample vectors on a high dimensional unit sphere and classify them based on inner product with estimated class mean. This model is easier to analyse theoretically than real-world deep learning examples. We generate the data and form the classifiers (which are our target models) as follows:

1. For each class, we first sample a true class mean $m_c$ on a high dimensional unit sphere that is orthogonal to all other true class means ($\forall i, j \in \{1, \ldots, C\} : m_i \perp m_j \vee i = j$).

2. We sample $2S$ vectors $x_c$ for each class. We assume that they are Gaussian distributed around the the true class mean $x_c \sim \mathcal{N}(m_c, s^2 I)$ where the $s^2$ is the in-class variance.

3. For each "target model" we randomly choose a subset of size $CS$ from all generated vectors and compute per-class means $r_c$.

4. The computed mean is used to classify sample $x$ by computing the inner product $\langle x, r_c \rangle$ as a metric of similarity.

The attacker has to infer which vectors have been used for training the classifier. Instead of utilising the logits (like in many image classification tasks), the attacker can use the inner products of a point with the cluster means. Since the inner product score follows a normal distribution, LiRA with infinitely many shadow models is the optimal attack by the Neyman-Pearson lemma (Neyman & Pearson, 1933), which states that the likelihood ratio test is the most powerful test for a given FPR.

This simplified model resembles a linear (Head) classifier often used in transfer learning when adapting to a new dataset. We also focus on the linear (Head) classifier in our empirical evaluation in Section 4. In the linear classifier, we find a matrix $W$ and biases $b$, to optimize the cross-entropy between the labels and logits $Wv + b$, where $v$ denotes the feature space representation of the data. In the simplified model, the rows of $W$ are replaced by the cluster means and we do not include the bias term in the classification.

Now, applying Lemma 1 to the simplified model yields the following result.

**Theorem 3** (Per-example LiRA power-law). *Fix a target example $(x, y_x)$. For the simplified model with arbitrary $C$ and infinitely many shadow models, the per-example LiRA vulnerability is given as*

$$\text{TPR}_{\text{LiRA}}(x) = \Phi\left(\Phi^{-1}(\text{FPR}_{\text{LiRA}}(x)) + \frac{\langle x, x - m_x \rangle}{\sqrt{S}s\|x\|}\right), \tag{18}$$

*where $m_x$ is the true mean of class $y_x$. In addition, for large $S$ we have*

$$\log(\text{TPR}_{\text{LiRA}}(x) - \text{FPR}_{\text{LiRA}}(x)) \approx -\frac{1}{2}\log S - \frac{1}{2}\Phi^{-1}(\text{FPR}_{\text{LiRA}}(x))^2 + \log\frac{\langle x, x - m_x \rangle}{\|x\|s\sqrt{2\pi}}. \tag{19}$$

*Proof.* See Appendix C.3. □

An immediate upper bound is obtained from Theorem 3 by the Cauchy-Schwarz inequality:

$$\log(\text{TPR}_{\text{LiRA}}(x) - \text{FPR}_{\text{LiRA}}(x)) \leq -\frac{1}{2}\log S - \frac{1}{2}\Phi^{-1}(\text{FPR}_{\text{LiRA}}(x))^2 + \log\frac{\|x - m_x\|}{s\sqrt{2\pi}}. \tag{20}$$

This implies that if $\|x - m_x\|$ is bounded, then the worst-case vulnerability is also bounded. Hence we can significantly reduce the MIA vulnerability of all examples in this non-DP setting by simply increasing the number of examples per class. Similarly, employing Lemma 2 and the simplified model, we obtain the following upper bound for RMIA performance.

**Theorem 4** (Per-example RMIA power-law). *Fix a target example $(x, y_x)$. For the simplified model with infinitely many shadow models, the per-example RMIA vulnerability is given as*

$$\text{TPR}_{\text{RMIA}}(x) \leq \Phi\left(\Phi^{-1}(\alpha) + \frac{\psi(x, C)}{\sqrt{S}s}\right), \tag{21}$$

*where $\alpha \geq \text{FPR}_{\text{RMIA}}(x)$ and*

$$\psi(x, C) =$$

$$\frac{\mathbb{E}_{(z,y_z)\sim\mathbb{D}}\left[2||x-z||^2 \mid y_z = y_x\right] + (C-1)\mathbb{E}_{(z,y_z)\sim\mathbb{D}}\left[(||x-m_x||^2 + ||z-m_z||^2) \mid y_z \neq y_x\right]}{\mathbb{E}_{(z,y_z)\sim\mathbb{D}}\left[2||x-z|| \mid y_z = y_x\right] + (C-1)\mathbb{E}_{(z,y_z)\sim\mathbb{D}}\left[(||x-m_x|| + ||z-m_z||) \mid y_z \neq y_x\right]}. \tag{22}$$

*In addition, for large $S$ we have*

$$\log(\text{TPR}_{\text{RMIA}}(x) - \text{FPR}_{\text{RMIA}}(x)) \leq -\frac{1}{2}\log S - \frac{1}{2}\Phi^{-1}(\alpha)^2 + \log\frac{\psi(x,C)}{\sqrt{2\pi}}. \tag{23}$$

*Proof.* See Appendix C.4. □

As for the LiRA power-law, bounding $||x - m_x||$ and $||z - m_z||$ will provide a worst-case upper bound for which the power-law holds. Now the following corollaries extend the power-law to the average-case MIA vulnerabilities. We will also empirically validate these results in Section 4.

**Corollary 5** (Average-case LiRA power-law). *For the simplified model with arbitrary $C$, sufficiently large $S$ and infinitely many shadow models, we have*

$$\log(\overline{\text{TPR}}_{\text{LiRA}} - \overline{\text{FPR}}_{\text{LiRA}}) \approx -\frac{1}{2}\log S - \frac{1}{2}\Phi^{-1}(\overline{\text{FPR}}_{\text{LiRA}})^2 + \log\left(\mathbb{E}_{(x,y_x)\sim\mathbb{D}}\left[\frac{\langle x, x - m_x\rangle}{\sqrt{2\pi}||x||s}\right]\right). \tag{24}$$

*Proof.* See Appendix C.5. □

**Corollary 6** (Average-case RMIA power-law). *For the simplified model with sufficiently large $S$ and infinitely many shadow models, we have*

$$\log(\overline{\text{TPR}}_{\text{RMIA}} - \overline{\text{FPR}}_{\text{RMIA}}) \leq -\frac{1}{2}\log S - \frac{1}{2}\Phi^{-1}(\alpha)^2 + \log\left(\mathbb{E}_{(x,y_x)\sim\mathbb{D}}\left[\frac{\psi(x,C)}{\sqrt{2\pi}}\right]\right). \tag{25}$$

*Proof.* See Appendix C.6. □

## 4 EMPIRICAL EVALUATION OF MIA VULNERABILITY AND DATASET PROPERTIES

In this section, we investigate how different properties of datasets affect the MIA vulnerability. Based on our observations, we propose a method to predict the vulnerability to MIA using these properties.

### 4.1 EXPERIMENTAL SETUP

We focus on a image classification setting where we fine-tune pre-trained models on sensitive downstream datasets and assess the MIA vulnerability using LiRA and RMIA with $M = 256$ shadow/reference models. We base our experiments on a subset of the few-shot benchmark VTAB (Zhai et al., 2019) that achieves a test classification accuracy $> 80\%$ (see Table A2).

We report results for fine-tuning a last layer classifier (Head) trained on top of a Vision Transformer ViT-Base-16 (ViT-B; Dosovitskiy et al., 2021), pre-trained on ImageNet-21k (Russakovsky et al., 2015). The results for using ResNet-50 (R-50; Kolesnikov et al., 2020) as a backbone can be found in Appendix F.1. We optimise the hyperparameters (batch size, learning rate and number of epochs) using the library Optuna (Akiba et al., 2019) with the Tree-structured Parzen Estimator (TPE; Bergstra et al., 2011) sampler with 20 iterations (more details in Appendix E.2). We provide the the code for reproducing the experiments in the supplementary material.

**Measuring the uncertainty for TPR** The TPR values from the LiRA-based classifier can be seen as maximum likelihood-estimators for the probability of producing true positives among the positive

samples. Since we have a finite number of samples for our estimation, it is important to estimate the uncertainty in these estimators. Therefore, when we report the TPR values for a single repeat of the learning algorithm, we estimate the stochasticity of the TPR estimate by using Clopper-Pearson intervals (Clopper & Pearson, 1934). Given TP true positives among P positives, the $1-\alpha$ confidence Clopper-Pearson interval for the TPR is given as

$$
\begin{aligned}
B(\alpha/2; \text{TP}, \text{P} - \text{TP} + 1) &< \text{TPR} \\
\text{TPR} &< B(1 - \alpha/2; \text{TP} + 1, \text{P} - \text{TP}),
\end{aligned}
\tag{26}
$$

where $B(q; a, b)$ is the $q$th-quantile of Beta$(a, b)$ distribution.

### 4.2 EXPERIMENTAL RESULTS

Using the setting described above, we study how the number of classes and the number of shots affect the vulnerability (TPR at FPR as described in Section 2) using LiRA. We make the following observations:

- A larger number of $S$ (**shots**) decrease the vulnerability in a power law relation as demonstrated in Figure 1a. We provide further evidence of this in the Appendix (Figure A.2 and Tables A3 and A4).
- Contrary, a larger number of $C$ (**classes**) increases the vulnerability as demonstrated in Figure 1b with further evidence in Figure A.3 and Tables A5 and A6 in the Appendix. However, the trend w.r.t. $C$ is not as clear as with $S$.

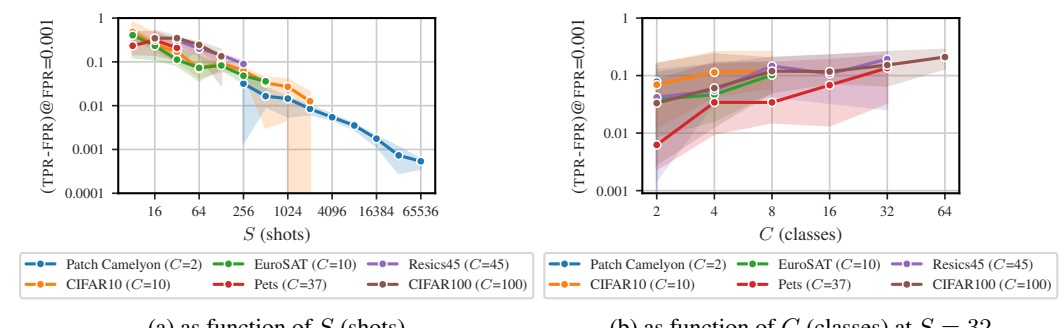

(a) as function of $S$ (shots)     (b) as function of $C$ (classes) at $S = 32$

Figure 1: LiRA vulnerability ((TPR $-$ FPR) at FPR $= 0.001$) as a function of dataset properties when attacking a ViT-B Head fine-tuned without DP on different datasets. We observe a power-law relation between the MIA vulnerability and $S$ (shots) in Figure 1a while the number of classes $C$ has a small effect on the MIA vulnerbility in Figure 1b. The solid line displays the median and the error bars the minimum of the lower bounds and maximum of the upper bounds for the Clopper-Pearson CIs over multiple seeds (six for Figure 1a and 12 for Figure 1b)

**RMIA** In Figure 2 we compare the vulnerability of the models to LiRA and RMIA as a function of the number of $S$ (shots) at FPR $= 0.1$. We observe the power-law for both attacks, but the RMIA is more unstable than LiRA (especially for lower FPR). More results for RMIA are in Figures A.6 to A.8 in the Appendix.

### 4.3 MODEL TO PREDICT DATASET VULNERABILITY

The trends seen in Figure 1 suggest the same power law relationship that we derived for the simplified model of membership inference in Section 3. We fit a linear regression model to predict $\log(\text{TPR} - \text{FPR})$ for each FPR $= 10^{-k}, k = 1, \ldots, 5$ separately using the $\log C$ and $\log S$ as covariates with statsmodels (Seabold & Perktold, 2010). The general form of the model can be found in Equation (27), where $\beta_S, \beta_C$ and $\beta_0$ are the learnable regression parameters.

$$
\log_{10}(\text{TPR} - \text{FPR}) = \beta_S \log_{10}(S) + \beta_C \log_{10}(C) + \beta_0
\tag{27}
$$

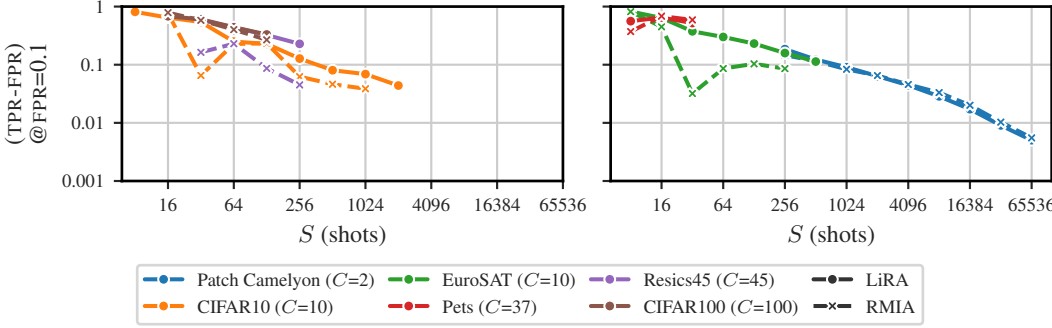

Figure 2: LiRA and RMIA vulnerability (($\text{TPR} - \text{FPR}$) at $\text{FPR} = 0.1$) as a function of shots ($S$) when attacking a ViT-B Head fine-tuned without DP on different datasets. For better visibility, we split the datasets into two panels. We observe the power-law for both attacks, but the RMIA is more unstable than LiRA. The lines display the median over six seeds.

In Appendix F.2, we propose a variation of the regression model that predicts $\log_{10}(\text{TPR})$ instead of $\log_{10}(\text{TPR} - \text{FPR})$ but this alternative model performs worse on our empirical data and predicts $\text{TPR} < \text{FPR}$ in the tail when $S$ is very large.

We utilise MIA results of ViT-B (Head) (see Table A3) as the training data. Based on the $R^2$ (coefficient of determination) score ($R^2 = 0.930$ for the model trained on $\text{FPR} = 0.001$ data), our model fits the data extremely well. We provide further evidence for other $\text{FPR}$ in Figure A.4 and Table A8 in the Appendix. Figure 3 shows the parameters of the prediction model fitted to the training data. For larger $\text{FPR}$, the coefficient $\beta_S$ is around $-0.5$, as our theoretical analysis predicts. However, the coefficient value decreases for small $\text{FPR}$. This is perhaps because the power-law in Equation (24) only holds for large $S$, and for small $\text{FPR}$ Equation (24) significantly underestimates the vulnerability in small-$S$ regime (see Appendix D).

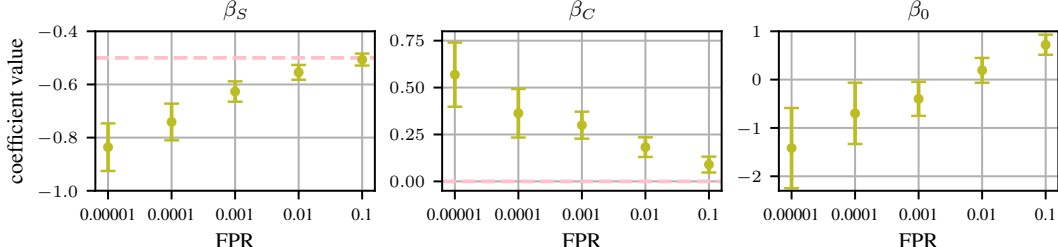

Figure 3: Coefficient values for different $\text{FPR}$ when fitting a regression model based on Equation (27) fitted on data from ViT-B (Head) with LiRA (Table A3). The error bars display the 95% confidence intervals based on Student's t-distribution. Theoretical values in the simplified model is shown by pink dotted lines ($\beta_S = 0.5$ and $\beta_C = 0$).

**Prediction quality on other MIA target models**   We analyse how the regression model trained on the ViT-B (Head) data generalizes to other target models. The main points are:

- *R-50 (Head):* Figure 4a shows that the regression model is robust to a change of the feature extractor, as it is able to predict the $\text{TPR}$ for R-50 (Head) (test $R^2 = 0.790$).

- *R-50 (FiLM):* Figure 4b shows that the prediction quality is good for R-50 (FiLM) models. These models are fine-tuned with parameter-efficient FiLM (Perez et al., 2018) layers (See Appendix E.1). Tobaben et al. (2023) demonstrated that FiLM layers are a competitive alternative to training all parameters. We supplement the MIA results of Tobaben et al. (2023) with own FiLM training runs. Refer to Table A7 in the Appendix.

- *From-Scratch-Training:* Carlini et al. (2022) provide limited results on from-scratch-training. To the best of our knowledge these are the only published LiRA results on image classification

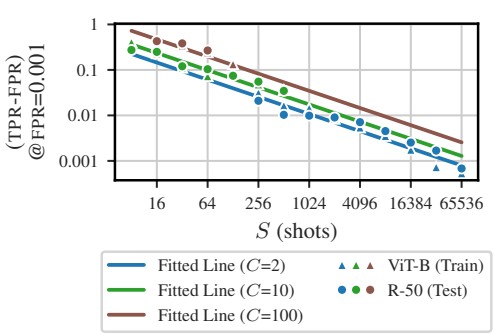

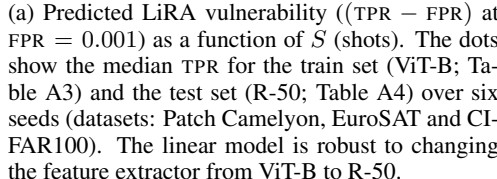

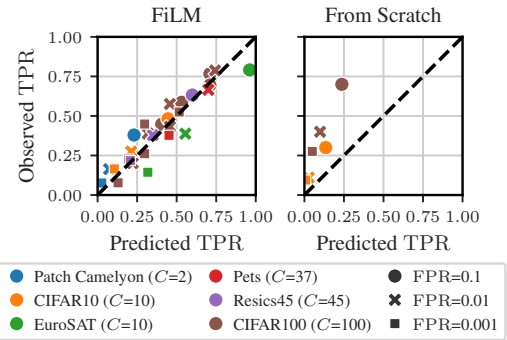

(a) Predicted LiRA vulnerability ((TPR − FPR) at FPR = 0.001) as a function of $S$ (shots). The dots show the median TPR for the train set (ViT-B; Table A3) and the test set (R-50; Table A4) over six seeds (datasets: Patch Camelyon, EuroSAT and CIFAR100). The linear model is robust to changing the feature extractor from ViT-B to R-50.

(b) Evaluating prediction performance for target models (i) left: fine-tuned with FiLM (data partially from Tobaben et al. (2023), see Table A7) (ii) right: trained from scratch (data from Carlini et al. (2022)). The linear model is robust to changing the fine-tuning method from Head to FiLM, but from scratch training seems to be more vulnerable than predicted. Note that the results from Carlini et al. (2022) use data augmentation while we do not.

Figure 4: Performance of the regression model based on Equation (27) fitted on data from Table A3.

models. Figure 4b displays that our prediction model underestimates the vulnerability of the from-scratch trained target models. We have identified two potential explanations for this (i) In from-scratch-training all weights of the model need to be trained from the sensitive data and thus potentially from-scratch-training could be more vulnerable than fine-tuning. (ii) The strongest attack in Carlini et al. (2022) uses data augmentations to improve the performance. We are not using this optimization.

## 5 DISCUSSION

Under the GDPR and similar legal regimes, machine learning (ML) models trained on personal data that memorise the data are personal data and need to be carefully protected. Our work analyses in which cases trained models would most likely be personal data and in which cases they might not be. This will help in evaluating the risk of different kinds of models, favouring less risky models when possible and paying extra attention to more risky cases.

As the best means of protecting privacy, differential privacy, reduces the utility of models, it is important to understand when it is necessary. Aligning with the prior literature, our results highlight that models are the most vulnerable to MIA when the number of examples per class is low. A key result of the present paper is, however, the power-law relationship. This has a potentially useful implication that a practitioner could reduce the MIA vulnerability and estimate how large a dataset would be needed to mitigate the vulnerability in the non-DP transfer learning setting. The practitioner could focus on the class with least examples, while taking into account that the number of classes would not be completely independent of the vulnerability.

One major reason for MIA vulnerability can be memorisation of the training data. Feldman & Zhang (2020) experimentally test memorisation in neural network training, and find that according to their definition, a large fraction of the training data are memorised when training from scratch, while only few are when fine-tuning. This is aligned with our results that indicate from scratch training to be more vulnerable than fine-tuning.

**Limitations** Despite the theoretical analysis on the optimal score-based MIA, the vulnerability to white-box attacks and future stronger attacks might behave differently. Also, our results assume well-behaved underlying distributions. Formal bounds on MIA vulnerability would require something like DP. In addition, both our theoretical and empirical analysis focus on deep transfer learning using fine-tuning. Models trained from scratch are likely to be more vulnerable.

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

## A   FORMULATING LiRA AND RMIA FOR SECTION 3

Let $\mathcal{M}$ be our target model and $\ell(\mathcal{M}(x), y_x)$ be the loss of the model on a target example $(x, y_x)$. The goal of MIA is to determine whether $(x, y_x) \in \mathcal{D}_{\text{target}}$. This can be formulated as a hypothesis test:

$$H_0 : (x, y_x) \notin \mathcal{D}_{\text{target}} \tag{A1}$$

$$H_1 : (x, y_x) \in \mathcal{D}_{\text{target}}. \tag{A2}$$

### A.1   LiRA

Following (Carlini et al., 2022), we formulate the Likelihood Ratio Attack (LiRA). LiRA exploits the difference of losses on the target model under $H_0$ and $H_1$. To model the IN/OUT loss distributions with few shadow models, LiRA employs a parametric modelling. Particularly, LiRA models $t_x$ by a normal distribution. That is, the hypothesis test formulated above can be rewritten as

$$H_0' : t_x \sim \mathcal{N}(\hat{\mu}_{\text{out}}, \hat{\sigma}_{\text{out}}) \tag{A3}$$

$$H_1' : t_x \sim \mathcal{N}(\hat{\mu}_{\text{in}}, \hat{\sigma}_{\text{in}}). \tag{A4}$$

The likelihood ratio is now

$$\text{LR}(x) = \frac{\mathcal{N}(t_x; \hat{\mu}_{\text{in}}, \hat{\sigma}_{\text{in}})}{\mathcal{N}(t_x; \hat{\mu}_{\text{out}}, \hat{\sigma}_{\text{out}})}. \tag{A5}$$

LiRA rejects $H_0'$ if and only if

$$\text{LR}(x) \geq \beta, \tag{A6}$$

concluding that $H_1'$ is true, i.e., identifying the membership of $(x, y_x)$. Thus, the true positive rate of this hypothesis test given as

$$\text{TPR}_{\text{LiRA}}(x) = \Pr_{\mathcal{D}_{\text{target}} \sim \mathbb{D}^{|\mathcal{D}|}, \phi^M} \left( \frac{\mathcal{N}(t_x; \hat{\mu}_{\text{in}}(x), \hat{\sigma}_{\text{in}}(x)^2)}{\mathcal{N}(t_x; \hat{\mu}_{\text{out}}(x), \hat{\sigma}_{\text{out}}(x)^2)} \geq \beta \mid (x, y_x) \in \mathcal{D}_{\text{target}} \right), \tag{A7}$$

where $\phi^M$ denotes the randomness in the shadow set sampling and shadow model training.

### A.2   RMIA

By modelling $t_z$ by a normal distribution, Zarifzadeh et al. (2024) approximate the *pairwise* likelihood ratio as

$$\text{LR}(x, z) = \frac{p(\theta \mid x, y_x)}{p(\theta \mid z, y_z)} \approx \frac{\mathcal{N}(t_x; \hat{\mu}_{x,x}, \hat{\sigma}_{x,x}^2)\mathcal{N}(t_z; \hat{\mu}_{x,z}, \hat{\sigma}_{x,z}^2)}{\mathcal{N}(t_x; \hat{\mu}_{z,x}, \hat{\sigma}_{z,x}^2)\mathcal{N}(t_z; \hat{\mu}_{z,z}, \hat{\sigma}_{z,z}^2)}, \tag{A8}$$

where $\hat{\mu}_{a,b}$ and $\hat{\sigma}_{a,b}$ are, respectively, the mean and standard deviation of $t_b$ estimated from shadow models when the training set contains $a$ but not $b$. Then RMIA exploits the probability of rejecting the pairwise likelihood ratio test over $(z, y_z) \sim \mathbb{D}$:

$$\text{Score}_{\text{RMIA}}(x) = \Pr_{(z, y_z) \sim \mathbb{D}} \left( \text{LR}(x, z) \geq \gamma \right). \tag{A9}$$

Thus, RMIA rejects $H_0$ if and only if

$$\Pr_{(z, y_z) \sim \mathbb{D}} \left( \text{LR}(x, z) \geq \gamma \right) \geq \beta, \tag{A10}$$

identifying the membership of $x$. Hence the true positive rate of RMIA is given as

$$
\text{TPR}_{\text{RMIA}}(x) = \Pr_{\mathcal{D}_{\text{target}} \sim \mathbb{D}^{|\mathcal{D}|}, \phi^M} \left( \Pr_{(z, y_z) \sim \mathbb{D}} (\text{LR}(x, z) \geq \gamma) \geq \beta \mid (x, y_x) \in \mathcal{D}_{\text{target}} \wedge (x, y_x) \notin \mathcal{D}_{\text{target}} \right),
$$

(A11)

where $\phi^M$ denotes the randomness in the sahdow set sampling and shadow model training.

# B   ON THE ASSUMPTIONS IN SECTION 3

## B.1   THE ASSUMPTION OF SHARED VARIANCES

In Section 3 we assumed that for LiRA $\sigma_{\text{in}} = \sigma_{\text{out}}$ and $\hat{\sigma}_{\text{in}} = \hat{\sigma}_{\text{out}}$, and that for RMIA $\sigma_x = \sigma_{x,x} = \sigma_{z,x}$, $\sigma_z = \sigma_{x.z} = \sigma_{z,z}$, $\hat{\sigma}_x = \hat{\sigma}_{x,x} = \hat{\sigma}_{z,x}$ and $\hat{\sigma}_z = \hat{\sigma}_{x.z} = \hat{\sigma}_{z,z}$. Utilising the simplified model formulated in Section 3.4, we show that for large enough number $S$ of examples per class these assumptions are reasonable.

From the proof of Theorem 3 (see Appendix C.3) we have

$$
\sigma_{\text{in}}^2 = \hat{\sigma}_{\text{in}}^2 = \text{Var}(s_{y_x}^{(\text{in})}) = \frac{1}{S} \left( 1 - \frac{1}{S} \right) ||x||^2 s^2
$$

(A12)

$$
\sigma_{\text{out}}^2 = \hat{\sigma}_{\text{out}}^2 = \text{Var}(s_{y_x}^{(\text{out})}) = \frac{1}{S} ||x||^2 s^2
$$

(A13)

Thus, the differences $\sigma_{\text{in}} - \sigma_{\text{out}}$ and $\hat{\sigma}_{\text{in}} - \hat{\sigma}_{\text{out}}$ are negligible for large $S$. Similarly, we have

$$
\sigma_{x,x}^2 = \hat{\sigma}_{x,x}^2 = \text{Var}(s_{y_x}^{(x)}(x)) = \frac{1}{S} \left( 1 - \frac{1}{S} \right) ||x||^2 s^2
$$

(A14)

$$
\sigma_{z,x}^2 = \hat{\sigma}_{z,x}^2 = \text{Var}(s_{y_x}^{(z)}(x)) = \begin{cases} \frac{1}{S} \left( 1 - \frac{1}{S} \right) ||x||^2 s^2 & \text{if } y_x = y_z \\ \frac{1}{S} ||x||^2 s^2 & \text{if } y_x \neq y_z \end{cases}
$$

(A15)

$$
\sigma_{x,z}^2 = \hat{\sigma}_{x,z}^2 = \text{Var}(s_{y_z}^{(x)}(z)) = \begin{cases} \frac{1}{S} \left( 1 - \frac{1}{S} \right) ||z||^2 s^2 & \text{if } y_x = y_z \\ \frac{1}{S} ||z||^2 s^2 & \text{if } y_x \neq y_z \end{cases}
$$

(A16)

$$
\sigma_{z,z}^2 = \hat{\sigma}_{z,z}^2 = \text{Var}(s_{y_z}^{(z)}(z)) = \frac{1}{S} \left( 1 - \frac{1}{S} \right) ||z||^2 s^2.
$$

(A17)

Therefore, the differences $\sigma_{x,x} - \sigma_{z,x}$, $\sigma_{x.z} - \sigma_{z,z}$, $\hat{\sigma}_{x,x} - \hat{\sigma}_{z,x}$ and $\hat{\sigma}_{x.z} - \hat{\sigma}_{z,z}$ are negligible for large enough $S$. Hence as long as the simplified model approximates classification tasks to which Lemmas 1 and 2 are applied, these assumptions are reasonably justified.

## B.2   RELAXING THE ASSUMPTION OF LEMMA 1

In Lemma 1 we assume that an attacker has access to the true underlying distribution. However, this is not necessarily the case in real-world settings. Noting that the Equation (12) mainly relies on the true location parameters $\mu_{\text{in}}(x)$, $\mu_{\text{out}}(x)$ and scale parameter $\sigma(x)$, we may relax this assumption.

Notice that if we completely drop this assumption so that an attacker trains shadow models with an arbitrary underlying distribution, then we may not be able to choose a desired $\text{FPR}_{\text{LiRA}}(x)$. From Equation (A34) in the proof of Lemma 1 we have

$$
\frac{\hat{\sigma}^2 \log \beta}{\sigma(\hat{\mu}_{\text{in}} - \hat{\mu}_{\text{out}})} + \frac{\hat{\mu}_{\text{in}} + \hat{\mu}_{\text{out}}}{2\sigma} - \frac{\mu_{\text{out}}}{\sigma} = \begin{cases} F_t^{-1}(1 - \text{FPR}_{\text{LiRA}}(x)) & \text{if } \hat{\mu}_{\text{in}} > \hat{\mu}_{\text{out}} \\ F_t^{-1}(\text{FPR}_{\text{LiRA}}(x)) & \text{if } \hat{\mu}_{\text{in}} < \hat{\mu}_{\text{out}} \end{cases}
$$

(A18)

$$
\log \beta = \begin{cases} \frac{\hat{\mu}_{\text{in}} - \hat{\mu}_{\text{out}}}{\hat{\sigma}^2} \left( \sigma F_t^{-1}(1 - \text{FPR}_{\text{LiRA}}(x)) - \frac{\hat{\mu}_{\text{in}} + \hat{\mu}_{\text{out}}}{2} + \mu_{\text{out}} \right) & \text{if } \hat{\mu}_{\text{in}} > \hat{\mu}_{\text{out}} \\ \frac{\hat{\mu}_{\text{in}} - \hat{\mu}_{\text{out}}}{\hat{\sigma}^2} \left( \sigma F_t^{-1}(\text{FPR}_{\text{LiRA}}(x)) - \frac{\hat{\mu}_{\text{in}} + \hat{\mu}_{\text{out}}}{2} + \mu_{\text{out}} \right) & \text{if } \hat{\mu}_{\text{in}} < \hat{\mu}_{\text{out}}, \end{cases}
$$

(A19)

where we abuse notations by denoting $\mu_{\text{in}}$ to refer to $\mu_{\text{in}}(x)$ and similarly for other parameters. Since we need to choose a rejection region of the likelihood ratio test such that $\beta \geq 1$, we have

$$
\begin{cases} \frac{\hat{\mu}_{\text{in}} - \hat{\mu}_{\text{out}}}{\hat{\sigma}^2} \left( \sigma F_t^{-1}(1 - \text{FPR}_{\text{LiRA}}(x)) - \frac{\hat{\mu}_{\text{in}} + \hat{\mu}_{\text{out}}}{2} + \mu_{\text{out}} \right) \geq 0 & \text{if } \hat{\mu}_{\text{in}} > \hat{\mu}_{\text{out}} \\ \frac{\hat{\mu}_{\text{in}} - \hat{\mu}_{\text{out}}}{\hat{\sigma}^2} \left( \sigma F_t^{-1}(\text{FPR}_{\text{LiRA}}(x)) - \frac{\hat{\mu}_{\text{in}} + \hat{\mu}_{\text{out}}}{2} + \mu_{\text{out}} \right) \geq 0 & \text{if } \hat{\mu}_{\text{in}} < \hat{\mu}_{\text{out}}. \end{cases}
$$

(A20)

Therefore, the sufficient condition about attacker's knowledge on the underlying distribution for Lemma 1 to hold is

$$\begin{cases} \sigma F_t^{-1}(1 - \text{FPR}_{\text{LiRA}}(x)) - \frac{\hat{\mu}_{\text{in}} + \hat{\mu}_{\text{out}}}{2} + \mu_{\text{out}} \geq 0 & \text{if } \hat{\mu}_{\text{in}} > \hat{\mu}_{\text{out}} \\ \sigma F_t^{-1}(\text{FPR}_{\text{LiRA}}(x)) - \frac{\hat{\mu}_{\text{in}} + \hat{\mu}_{\text{out}}}{2} + \mu_{\text{out}} \leq 0 & \text{if } \hat{\mu}_{\text{in}} < \hat{\mu}_{\text{out}}. \end{cases} \tag{A21}$$

We summarise this discussion in the following:

**Lemma A1** (Lemma 1 with a relaxed assumption). *Suppose that the true distribution of $t_x$ is of location-scale family with locations $\mu_{\text{in}}(x), \mu_{\text{out}}(x)$ and scale $\sigma(x)$, and that LiRA models $t_x$ by $\mathcal{N}(\hat{\mu}_{\text{in}}(x), \hat{\sigma}(x))$ and $\mathcal{N}(\hat{\mu}_{\text{out}}(x), \hat{\sigma}(x))$. Assume that an attacker estimates parameters $\hat{\mu}_{\text{in}}(x), \hat{\sigma}(x), \hat{\mu}_{\text{out}}(x)$ and $\hat{\sigma}_{\text{out}}(x)$ with an approximated underlying distribution such that*

$$\begin{cases} \sigma F_t^{-1}(1 - \text{FPR}_{\text{LiRA}}(x)) - \frac{\hat{\mu}_{\text{in}} + \hat{\mu}_{\text{out}}}{2} + \mu_{\text{out}} \geq 0 & \text{if } \hat{\mu}_{\text{in}} > \hat{\mu}_{\text{out}} \\ \sigma F_t^{-1}(\text{FPR}_{\text{LiRA}}(x)) - \frac{\hat{\mu}_{\text{in}} + \hat{\mu}_{\text{out}}}{2} + \mu_{\text{out}} \leq 0 & \text{if } \hat{\mu}_{\text{in}} < \hat{\mu}_{\text{out}}. \end{cases} \tag{A22}$$

*Then the LiRA vulnerability of a fixed target example is*

$$\text{TPR}_{\text{LiRA}}(x) = \begin{cases} 1 - F_t\left(F_t^{-1}(1 - \text{FPR}_{\text{LiRA}}(x)) - \frac{\mu_{\text{in}}(x) - \mu_{\text{out}}(x)}{\sigma(x)}\right) & \text{if } \hat{\mu}_{\text{in}}(x) > \hat{\mu}_{\text{out}}(x) \\ F_t\left(F_t^{-1}(\text{FPR}_{\text{LiRA}}(x)) - \frac{\mu_{\text{in}}(x) - \mu_{\text{out}}(x)}{\sigma(x)}\right) & \text{if } \hat{\mu}_{\text{in}}(x) < \hat{\mu}_{\text{out}}(x), \end{cases} \tag{A23}$$

*where $F_t$ is the cdf of $t$ with the standard location and unit scale.*

## C  MISSING PROOFS OF SECTION 3

### C.1  PROOF OF LEMMA 1

**Lemma 1** (Per-example LiRA vulnerability). *Suppose that the true distribution of $t_x$ is of location-scale family with locations $\mu_{\text{in}}(x), \mu_{\text{out}}(x)$ and scale $\sigma(x)$, and that LiRA models $t_x$ by $\mathcal{N}(\hat{\mu}_{\text{in}}(x), \hat{\sigma}(x))$ and $\mathcal{N}(\hat{\mu}_{\text{out}}(x), \hat{\sigma}(x))$. Assume that an attacker has access to the underlying distribution $\mathbb{D}$. Then for a large enough number of examples per class and infinitely many shadow models, the LiRA vulnerability of a fixed target example is*

$$\text{TPR}_{\text{LiRA}}(x) = \begin{cases} 1 - F_t\left(F_t^{-1}(1 - \text{FPR}_{\text{LiRA}}(x)) - \frac{\mu_{\text{in}}(x) - \mu_{\text{out}}(x)}{\sigma(x)}\right) & \text{if } \hat{\mu}_{\text{in}}(x) > \hat{\mu}_{\text{out}}(x) \\ F_t\left(F_t^{-1}(\text{FPR}_{\text{LiRA}}(x)) - \frac{\mu_{\text{in}}(x) - \mu_{\text{out}}(x)}{\sigma(x)}\right) & \text{if } \hat{\mu}_{\text{in}}(x) < \hat{\mu}_{\text{out}}(x), \end{cases} \tag{12}$$

*where $F_t$ is the cdf of $t$ with the standard location and unit scale.*

*Proof.* We abuse notations by denoting $\mu_{\text{in}}$ to refer to $\mu_{\text{in}}(x)$ and similarly for other statistics. We have

$$\log \frac{\mathcal{N}(t_x; \hat{\mu}_{\text{in}}, \hat{\sigma})}{\mathcal{N}(t_x; \hat{\mu}_{\text{out}}, \hat{\sigma})} \geq \log \beta \tag{A24}$$

$$-\frac{1}{2}\left(\frac{t_x - \hat{\mu}_{\text{in}}}{\hat{\sigma}}\right)^2 + \frac{1}{2}\left(\frac{t_x - \hat{\mu}_{\text{out}}}{\hat{\sigma}}\right)^2 \geq \log \beta \tag{A25}$$

$$\frac{1}{2\hat{\sigma}^2}(2t_x\hat{\mu}_{\text{in}} - \hat{\mu}_{\text{in}}^2 - 2t_x\hat{\mu}_{\text{out}} + \hat{\mu}_{\text{out}}^2) \geq \log \beta \tag{A26}$$

$$\frac{1}{2\hat{\sigma}^2}(\hat{\mu}_{\text{in}} - \hat{\mu}_{\text{out}})(2t_x - \hat{\mu}_{\text{in}} - \hat{\mu}_{\text{out}}) \geq \log \beta \tag{A27}$$

$$\begin{cases} t_x \geq \frac{\hat{\sigma}^2 \log \beta}{\hat{\mu}_{\text{in}} - \hat{\mu}_{\text{out}}} + \frac{\hat{\mu}_{\text{in}} + \hat{\mu}_{\text{out}}}{2} & \text{if } \hat{\mu}_{\text{in}} > \hat{\mu}_{\text{out}} \\ t_x \leq \frac{\hat{\sigma}^2 \log \beta}{\hat{\mu}_{\text{in}} - \hat{\mu}_{\text{out}}} + \frac{\hat{\mu}_{\text{in}} + \hat{\mu}_{\text{out}}}{2} & \text{if } \hat{\mu}_{\text{in}} < \hat{\mu}_{\text{out}}. \end{cases} \tag{A28}$$

Then if $\hat{\mu}_{\text{in}} > \hat{\mu}_{\text{out}}$, in the limit of infinitely many shadow models

$$\text{FPR}_{\text{LiRA}}(x) = \Pr_t\left(\mu_{\text{out}} + \sigma t \geq \frac{\hat{\sigma}^2 \log \beta}{\hat{\mu}_{\text{in}} - \hat{\mu}_{\text{out}}} + \frac{\hat{\mu}_{\text{in}} + \hat{\mu}_{\text{out}}}{2}\right) \tag{A29}$$

$$= \Pr_t\left(t \geq \frac{\hat{\sigma}^2 \log \beta}{\sigma(\hat{\mu}_{\text{in}} - \hat{\mu}_{\text{out}})} + \frac{\hat{\mu}_{\text{in}} + \hat{\mu}_{\text{out}}}{2\sigma} - \frac{\mu_{\text{out}}}{\sigma}\right) \tag{A30}$$

$$= 1 - F_t\left(\frac{\hat{\sigma}^2 \log \beta}{\sigma(\hat{\mu}_{\text{in}} - \hat{\mu}_{\text{out}})} + \frac{\hat{\mu}_{\text{in}} + \hat{\mu}_{\text{out}}}{2\sigma} - \frac{\mu_{\text{out}}}{\sigma}\right), \tag{A31}$$

and if $\hat{\mu}_{\text{in}} < \hat{\mu}_{\text{out}}$, similarly,

$$\text{FPR}_{\text{LiRA}}(x) = \Pr_t \left( \mu_{\text{out}} + \sigma t \leq \frac{\hat{\sigma}^2 \log \beta}{\hat{\mu}_{\text{in}} - \hat{\mu}_{\text{out}}} + \frac{\hat{\mu}_{\text{in}} + \hat{\mu}_{\text{out}}}{2} \right) \tag{A32}$$

$$= F_t \left( \frac{\hat{\sigma}^2 \log \beta}{\sigma(\hat{\mu}_{\text{in}} - \hat{\mu}_{\text{out}})} + \frac{\hat{\mu}_{\text{in}} + \hat{\mu}_{\text{out}}}{2\sigma} - \frac{\mu_{\text{out}}}{\sigma} \right). \tag{A33}$$

Thus

$$\frac{\hat{\sigma}^2 \log \beta}{\sigma(\hat{\mu}_{\text{in}} - \hat{\mu}_{\text{out}})} + \frac{\hat{\mu}_{\text{in}} + \hat{\mu}_{\text{out}}}{2\sigma} - \frac{\mu_{\text{out}}}{\sigma} = \begin{cases} F_t^{-1}(1 - \text{FPR}_{\text{LiRA}}(x)) & \text{if } \hat{\mu}_{\text{in}} > \hat{\mu}_{\text{out}} \\ F_t^{-1}(\text{FPR}_{\text{LiRA}}(x)) & \text{if } \hat{\mu}_{\text{in}} < \hat{\mu}_{\text{out}}. \end{cases} \tag{A34}$$

It follows that if $\hat{\mu}_{\text{in}} > \hat{\mu}_{\text{out}}$,

$$\text{TPR}_{\text{LiRA}}(x) = \Pr_t \left( \mu_{\text{in}} + \sigma t \geq \frac{\hat{\sigma}^2 \log \beta}{\hat{\mu}_{\text{in}} - \hat{\mu}_{\text{out}}} + \frac{\hat{\mu}_{\text{in}} + \hat{\mu}_{\text{out}}}{2} \right) \tag{A35}$$

$$= \Pr_t \left( t \geq \frac{\hat{\sigma}^2 \log \beta}{\sigma(\hat{\mu}_{\text{in}} - \hat{\mu}_{\text{out}})} + \frac{\hat{\mu}_{\text{in}} + \hat{\mu}_{\text{out}}}{2\sigma} - \frac{\mu_{\text{in}}}{\sigma} \right) \tag{A36}$$

$$= 1 - F_t \left( F_t^{-1}(1 - \text{FPR}_{\text{LiRA}}(x)) - \frac{\mu_{\text{in}} - \mu_{\text{out}}}{\sigma} \right). \tag{A37}$$

If $\hat{\mu}_{\text{in}} < \hat{\mu}_{\text{out}}$, then

$$\text{TPR}_{\text{LiRA}}(x) = \Pr_t \left( \mu_{\text{in}} + \sigma t \leq \frac{\hat{\sigma}^2 \log \beta}{\hat{\mu}_{\text{in}} - \hat{\mu}_{\text{out}}} + \frac{\hat{\mu}_{\text{in}} + \hat{\mu}_{\text{out}}}{2} \right) \tag{A38}$$

$$= \Pr_t \left( t \leq \frac{\hat{\sigma}^2 \log \beta}{\sigma(\hat{\mu}_{\text{in}} - \hat{\mu}_{\text{out}})} + \frac{\hat{\mu}_{\text{in}} + \hat{\mu}_{\text{out}}}{2\sigma} - \frac{\mu_{\text{in}}}{\sigma} \right) \tag{A39}$$

$$= F_t \left( F_t^{-1}(\text{FPR}_{\text{LiRA}}(x)) - \frac{\mu_{\text{in}} - \mu_{\text{out}}}{\sigma} \right). \tag{A40}$$

$$\square$$

## C.2  PROOF OF LEMMA 2

**Lemma 2** (Per-example RMIA vulnerability). *Suppose that the true distributions of $t_x$ and $t_z$ are of location-scale family with locations $\mu_{x,x}, \mu_{z,x}, \mu_{x,z}, \mu_{z,z}$ and scales $\sigma_x, \sigma_z$, and that RMIA models $t_x$ and $t_z$ by normal distributions with parameters computed from shadow models. For a large enough number of examples per class and infinitely many shadow models, the RMIA vulnerability of a fixed target example is bounded by*

$$\text{TPR}_{\text{RMIA}}(x) \leq \begin{cases} 1 - F_t \left( F_t^{-1}(1 - \alpha) - \frac{\mathbb{E}_{(z,y_z)\sim\mathbb{D}}[q]}{\mathbb{E}_{(z,y_z)\sim\mathbb{D}}[A]} \right) & \text{if } \mathbb{E}_{(z,y_z)\sim\mathbb{D}}[A] > 0 \\ F_t \left( F_t^{-1}(\alpha) - \frac{\mathbb{E}_{(z,y_z)\sim\mathbb{D}}[q]}{\mathbb{E}_{(z,y_z)\sim\mathbb{D}}[A]} \right) & \text{if } \mathbb{E}_{(z,y_z)\sim\mathbb{D}}[A] < 0, \end{cases} \tag{15}$$

*for some constant $\alpha \geq \text{FPR}_{\text{RMIA}}(x)$, where*

$$q = \frac{(\mu_{x,x} - \mu_{z,x})(\hat{\mu}_{x,x} - \hat{\mu}_{z,x})}{\hat{\sigma}_x^2} - \frac{(\mu_{x,z} - \mu_{z,z})(\hat{\mu}_{x,z} - \hat{\mu}_{z,z})}{\hat{\sigma}_z^2} \tag{16}$$

$$A = \frac{\sigma_x}{\hat{\sigma}_x^2}(\hat{\mu}_{x,x} - \hat{\mu}_{z,x}) + \frac{\sigma_z}{\hat{\sigma}_z^2}(\hat{\mu}_{x,z} - \hat{\mu}_{z,z}). \tag{17}$$

*Proof.* We have

$$\text{LR}(x, z) \geq \gamma \tag{A41}$$

$$\frac{\exp\left(-\frac{1}{2}\left(\frac{t_x - \hat{\mu}_{x,x}}{\hat{\sigma}_x}\right)^2 - \frac{1}{2}\left(\frac{t_z - \hat{\mu}_{x,z}}{\hat{\sigma}_z}\right)^2\right)}{\exp\left(-\frac{1}{2}\left(\frac{t_x - \hat{\mu}_{z,x}}{\hat{\sigma}_x}\right)^2 - \frac{1}{2}\left(\frac{t_z - \hat{\mu}_{z,z}}{\hat{\sigma}_z}\right)^2\right)} \geq \gamma \tag{A42}$$

$$-\frac{1}{2}\left(\frac{t_x - \hat{\mu}_{x,x}}{\hat{\sigma}_x}\right)^2 + \frac{1}{2}\left(\frac{t_x - \hat{\mu}_{z,x}}{\hat{\sigma}_x}\right)^2$$

$$-\frac{1}{2}\left(\frac{t_z - \hat{\mu}_{x,z}}{\hat{\sigma}_z}\right)^2 + \frac{1}{2}\left(\frac{t_z - \hat{\mu}_{z,z}}{\hat{\sigma}_z}\right)^2 \geq \log\gamma \quad \text{(A43)}$$

$$\frac{1}{2\hat{\sigma}_x^2}(2t_x\hat{\mu}_{x,x} - \hat{\mu}_{x,x}^2 - 2t_x\hat{\mu}_{z,x} + \hat{\mu}_{z,x}^2)$$

$$+\frac{1}{2\hat{\sigma}_z^2}(2t_z\hat{\mu}_{x,z} - \hat{\mu}_{x,z}^2 - 2t_z\hat{\mu}_{z,z} + \hat{\mu}_{z,z}^2) \geq \log\gamma \quad \text{(A44)}$$

$$\frac{\hat{\mu}_{x,x} - \hat{\mu}_{z,x}}{2\hat{\sigma}_x^2}(2t_x - \hat{\mu}_{x,x} - \hat{\mu}_{z,x}) + \frac{\hat{\mu}_{x,z} - \hat{\mu}_{z,z}}{2\hat{\sigma}_z^2}(2t_z - \hat{\mu}_{x,z} - \hat{\mu}_{z,z}) \geq \log\gamma. \quad \text{(A45)}$$

When $(x, y_x) \in \mathcal{D}_{\text{target}}$ and $(z, y_z) \notin \mathcal{D}_{\text{target}}$, denote the left hand side of Equation (A45) by $\lambda_x$:

$$\lambda_x = \frac{\hat{\mu}_{x,x} - \hat{\mu}_{z,x}}{2\hat{\sigma}_x^2}(2\mu_{x,x} + 2\sigma_x t - \hat{\mu}_{x,x} - \hat{\mu}_{z,x}) \quad \text{(A46)}$$

$$+\frac{\hat{\mu}_{x,z} - \hat{\mu}_{z,z}}{2\hat{\sigma}_z^2}(2\mu_{x,z} + 2\sigma_z t - \hat{\mu}_{x,z} - \hat{\mu}_{z,z}). \quad \text{(A47)}$$

Similarly, when $(x, y_x) \notin \mathcal{D}_{\text{target}}$ and $(z, y_z) \in \mathcal{D}_{\text{target}}$, denoting the left hand side of Equation (A45) by $\lambda_z$, we have

$$\lambda_z = \frac{\hat{\mu}_{x,x} - \hat{\mu}_{z,x}}{2\hat{\sigma}_x^2}(2\mu_{z,x} + 2\sigma_x t - \hat{\mu}_{x,x} - \hat{\mu}_{z,x}) \quad \text{(A48)}$$

$$+\frac{\hat{\mu}_{x,z} - \hat{\mu}_{z,z}}{2\hat{\sigma}_z^2}(2\mu_{z,z} + 2\sigma_z t - \hat{\mu}_{x,z} - \hat{\mu}_{z,z}) \quad \text{(A49)}$$

$$=\left(\underbrace{\frac{\sigma_x}{\hat{\sigma}_x^2}(\hat{\mu}_{x,x} - \hat{\mu}_{z,x}) + \frac{\sigma_z}{\hat{\sigma}_z^2}(\hat{\mu}_{x,z} - \hat{\mu}_{z,z})}_{A}\right)t \quad \text{(A50)}$$

$$+\underbrace{\frac{\hat{\mu}_{x,x} - \hat{\mu}_{z,x}}{2\hat{\sigma}_x^2}(2\mu_{z,x} - \hat{\mu}_{x,x} - \hat{\mu}_{z,x}) + \frac{\hat{\mu}_{x,z} - \hat{\mu}_{z,z}}{2\hat{\sigma}_z^2}(2\mu_{z,z} - \hat{\mu}_{x,z} - \hat{\mu}_{z,z})}_{B} \quad \text{(A51)}$$

$$=At + B. \quad \text{(A52)}$$

Notice that $A$ and $B$ are functions of $z$ and independent of $t$. Thus $\mathbb{E}_{(z,y_z)\sim\mathbb{D}}[A]$ and $\mathbb{E}_{(z,y_z)\sim\mathbb{D}}[B]$ will be constants. We abuse notations by denoting $\mathbb{E}_z$ and $\text{Pr}_z$ to mean $\mathbb{E}_{(z,y_z)\sim\mathbb{D}}$ and $\text{Pr}_{(z,y_z)\sim\mathbb{D}}$, respectively. Note that taking probability over $t$ corresponds to calculating probability over sampling of the rest of the dataset other than the target example. By Markov's inequality, in the limit of infinitely many shadow models we have

$$\text{FPR}_{\text{RMIA}}(x) = \Pr_t\left(\Pr_z(e^{\lambda_z} \geq \gamma) \geq \beta\right) \leq \Pr_t\left(\frac{\mathbb{E}_z[e^{\lambda_z}]}{\gamma} \geq \beta\right) = \Pr_t\left(\mathbb{E}_z[e^{\lambda_z}] \geq \gamma\beta\right) \quad \text{(A53)}$$

Assuming that $\lambda_x$ and $\lambda_z$ have finite second moments, we can choose $\rho > 0$ such that

$$\mathbb{E}_z[e^{\lambda_x}] - e^{\mathbb{E}_z[\lambda_x]} \leq \rho \quad \text{(A54)}$$

$$\mathbb{E}_z[e^{\lambda_z}] - e^{\mathbb{E}_z[\lambda_z]} \leq \rho \quad \text{(A55)}$$

and $\rho$ is almost independent of $t$. Noting that $F_t^{-1}$ is an increasing function, we have

$$\text{FPR}_{\text{RMIA}}(x) \leq \Pr_t\left(\mathbb{E}_z[e^{\lambda_z}] \geq \gamma\beta\right) \quad \text{(A56)}$$

$$\leq \Pr_t\left(e^{\mathbb{E}_z[\lambda_z]} + \rho \geq \gamma\beta\right) \quad \text{(A57)}$$

$$= \Pr_t\left(\mathbb{E}_z[\lambda_z] \geq \log(\gamma\beta - \rho)\right) \quad \text{(A58)}$$

$$= \Pr_t\left(\mathbb{E}_z[A]t \geq \log(\gamma\beta - \rho) - \mathbb{E}_z[B]\right). \quad \text{(A59)}$$

Here we assume that $\gamma\beta > \rho$. Thus, assuming that $\mathbb{E}_z[A] \neq 0$, we can upper-bound $\text{FPR}_{\text{RMIA}}(x) \leq \alpha$ by setting

$$\alpha = \begin{cases} 1 - F_t\left(\frac{\log(\gamma\beta-\rho)-\mathbb{E}_z[B]}{\mathbb{E}_z[A]}\right) & \text{if } \mathbb{E}_z[A] > 0 \\ F_t\left(\frac{\log(\gamma\beta-\rho)-\mathbb{E}_z[B]}{\mathbb{E}_z[A]}\right) & \text{if } \mathbb{E}_z[A] < 0 \end{cases} \tag{A60}$$

That is,

$$\frac{\log(\gamma\beta-\rho)-\mathbb{E}_z[B]}{\mathbb{E}_z[A]} = \begin{cases} F_t^{-1}(1-\alpha) & \text{if } \mathbb{E}_z[A] > 0 \\ F_t^{-1}(\alpha) & \text{if } \mathbb{E}_z[A] < 0 \end{cases} \tag{A61}$$

Now let

$$q = \lambda_x - \lambda_z = \frac{(\mu_{x,x}-\mu_{z,x})(\hat{\mu}_{x,x}-\hat{\mu}_{z,x})}{\hat{\sigma}_x^2} + \frac{(\mu_{x,z}-\mu_{z,z})(\hat{\mu}_{x,z}-\hat{\mu}_{z,z})}{\hat{\sigma}_z^2}. \tag{A62}$$

Note that $q$ is also independent of $t$, thereby $\mathbb{E}_z[q]$ being a constant. By Markov's inequality, it follows that

$$\text{TPR}_{\text{RMIA}}(x) = \Pr_t\left(\Pr_z(e^{\lambda_x} \geq \gamma) \geq \beta\right) \tag{A63}$$

$$= \Pr_t\left(\Pr_z(e^{\lambda_z+q} \geq \gamma) \geq \beta\right) \tag{A64}$$

$$\leq \Pr_t\left(\frac{\mathbb{E}_z[e^{\lambda_z+q}]}{\gamma} \geq \beta\right) \tag{A65}$$

$$\leq \Pr_t\left(e^{\mathbb{E}_z[\lambda_z+q]} + \rho \geq \beta\gamma\right) \tag{A66}$$

$$= \Pr_t\left(\mathbb{E}_z[\lambda_z+q] \geq \log(\beta\gamma-\rho)\right) \tag{A67}$$

$$= \Pr_t\left(\mathbb{E}_z[A]t \geq \log(\beta\gamma-\rho) - \mathbb{E}_z[B] - \mathbb{E}_z[q]\right) \tag{A68}$$

$$= \begin{cases} \Pr_t\left(t \geq \frac{\log(\beta\gamma-\rho)-\mathbb{E}_z[B]}{\mathbb{E}_z[A]} - \frac{\mathbb{E}_z[q]}{\mathbb{E}_z[A]}\right) & \text{if } \mathbb{E}_z[A] > 0 \\ \Pr_t\left(t \leq \frac{\log(\beta\gamma-\rho)-\mathbb{E}_z[B]}{\mathbb{E}_z[A]} - \frac{\mathbb{E}_z[q]}{\mathbb{E}_z[A]}\right) & \text{if } \mathbb{E}_z[A] < 0. \end{cases} \tag{A69}$$

Hence we obtain

$$\text{TPR}_{\text{RMIA}}(x) = \begin{cases} 1 - F_t^{-1}\left(F_t^{-1}(1-\alpha) - \frac{\mathbb{E}_z[q]}{\mathbb{E}_z[A]}\right) & \text{if } \mathbb{E}_z[A] > 0 \\ F_t\left(F_t^{-1}(\alpha) - \frac{\mathbb{E}_z[q]}{\mathbb{E}_z[A]}\right) & \text{if } \mathbb{E}_z[A] < 0. \end{cases} \tag{A70}$$

$\square$

### C.3 PROOF OF THEOREM 3

**Theorem 3** (Per-example LiRA power-law). *Fix a target example $(x, y_x)$. For the simplified model with arbitrary $C$ and infinitely many shadow models, the per-example LiRA vulnerability is given as*

$$\text{TPR}_{\text{LiRA}}(x) = \Phi\left(\Phi^{-1}(\text{FPR}_{\text{LiRA}}(x)) + \frac{\langle x, x - m_x\rangle}{\sqrt{S}s||x||}\right), \tag{18}$$

*where $m_x$ is the true mean of class $y_x$. In addition, for large $S$ we have*

$$\log(\text{TPR}_{\text{LiRA}}(x) - \text{FPR}_{\text{LiRA}}(x)) \approx -\frac{1}{2}\log S - \frac{1}{2}\Phi^{-1}(\text{FPR}_{\text{LiRA}}(x))^2 + \log\frac{\langle x, x - m_x\rangle}{||x||s\sqrt{2\pi}}. \tag{19}$$

*Proof.* Let $\mathcal{D}_{\text{target}} = \{(x_{j,1}, j), ..., (x_{j,S}, j)\}_{j=1}^C$. Then the LiRA score of the target $(x, y_x)$ is

$$s_{y_x}^{(\text{in})} = \langle x, \frac{1}{S}\left(\sum_{i=1}^{S-1} x_{y_x,i} + x\right)\rangle = \langle x, \frac{1}{S}\sum_{i=1}^{S} x_{y_x,i}\rangle + \langle x, \frac{1}{S}(x - x_{y_x,S})\rangle \tag{A71}$$

$$s_{y_x}^{(\text{out})} = \langle x, \frac{1}{S}\sum_{i=1}^{S} x_{y_x,i}\rangle, \tag{A72}$$

respectively, when $(x, y_x) \in \mathcal{D}_{\text{target}}$ and when $(x, y_x) \notin \mathcal{D}_{\text{target}}$. Thus we obtain

$$\mu_{\text{in}} - \mu_{\text{out}} = \mathbb{E}[s_{y_x}^{(\text{in})} - s_{y_x}^{(\text{out})}] = \frac{1}{S}\langle x, x - m_x \rangle \tag{A73}$$

$$\sigma^2 = \text{Var}(s_{y_x}^{(\text{out})}) = \frac{1}{S}\text{Var}(\langle x, x_{y_x, i} \rangle) = \frac{1}{S}||x||^2 s^2 \tag{A74}$$

Noting that the LiRA score follows a normal distribution, by Lemma 1 we have

$$\text{TPR}_{\text{LiRA}}(x) = 1 - \Phi\left(\Phi^{-1}(1 - \text{FPR}_{\text{LiRA}}(x)) - \frac{\langle x, x - m_x \rangle}{\sqrt{S}s||x||}\right) \tag{A75}$$

$$= \Phi\left(\Phi^{-1}(\text{FPR}_{\text{LiRA}}(x)) + \frac{\langle x, x - m_x \rangle}{\sqrt{S}s||x||}\right), \tag{A76}$$

where $\Phi$ is the cdf of the standard normal distribution. This completes the first half of the theorem. Now we have

$$\text{TPR}_{\text{LiRA}}(x) = \Pr\left(\eta \leq \gamma_0 + \frac{\langle x, x - m_x \rangle}{\sqrt{S}s||x||}\right), \tag{A77}$$

$$\text{FPR}_{\text{LiRA}}(x) = \Pr(\eta \leq \gamma_0), \tag{A78}$$

where $\gamma_0$ is a tunable constant and $\eta \sim \mathcal{N}(0, 1)$. Thus for large enough $S$ we have

$$\text{TPR}_{\text{LiRA}}(x) - \text{FPR}_{\text{LiRA}}(x) \approx p_\eta(\gamma_0)\frac{\langle x, x - m_x \rangle}{\sqrt{S}||x||s} \tag{A79}$$

$$= \frac{1}{\sqrt{2\pi}}e^{-\frac{1}{2}\Phi^{-1}(\text{FPR}_{\text{LiRA}}(x))^2}\frac{\langle x, x - m_x \rangle}{\sqrt{S}||x||s}, \tag{A80}$$

$$\log(\text{TPR}_{\text{LiRA}}(x) - \text{FPR}_{\text{LiRA}}(x)) \approx -\frac{1}{2}\log S - \frac{1}{2}\Phi^{-1}(\text{FPR}_{\text{LiRA}}(x))^2 + \log\frac{\langle x, x - m_x \rangle}{||x||s\sqrt{2\pi}}. \tag{A81}$$

$$\square$$

## C.4 PROOF OF THEOREM 4

**Theorem 4** (Per-example RMIA power-law). *Fix a target example $(x, y_x)$. For the simplified model with infinitely many shadow models, the per-example RMIA vulnerability is given as*

$$\text{TPR}_{\text{RMIA}}(x) \leq \Phi\left(\Phi^{-1}(\alpha) + \frac{\psi(x, C)}{\sqrt{S}s}\right), \tag{21}$$

*where $\alpha \geq \text{FPR}_{\text{RMIA}}(x)$ and*

$$\psi(x, C) =$$
$$\frac{\mathbb{E}_{(z, y_z) \sim \mathbb{D}}\left[2||x - z||^2 \mid y_z = y_x\right] + (C - 1)\mathbb{E}_{(z, y_z) \sim \mathbb{D}}\left[(||x - m_x||^2 + ||z - m_z||^2) \mid y_z \neq y_x\right]}{\mathbb{E}_{(z, y_z) \sim \mathbb{D}}\left[2||x - z|| \mid y_z = y_x\right] + (C - 1)\mathbb{E}_{(z, y_z) \sim \mathbb{D}}\left[(||x - m_x|| + ||z - m_z||) \mid y_z \neq y_x\right]}. \tag{22}$$

*In addition, for large $S$ we have*

$$\log(\text{TPR}_{\text{RMIA}}(x) - \text{FPR}_{\text{RMIA}}(x)) \leq -\frac{1}{2}\log S - \frac{1}{2}\Phi^{-1}(\alpha)^2 + \log\frac{\psi(x, C)}{\sqrt{2\pi}}. \tag{23}$$

*Proof.* To apply Lemma 2, we will calculate $\mathbb{E}_z[q]$ and $\mathbb{E}_z[A]$. Let $s_{y_x}^{(x)}(x)$ (resp. $s_{y_x}^{(z)}(x)$) denote the score of the target $x$ for class $y_x$ when the dataset contains $(x, y_x)$ but not $(z, y_z)$ (resp. when the dataset contains $(z, y_z)$ but not $(x, y_x)$). Let $s_{y_z}^{(x)}(z)$ and $z_{y_z}^{(z)}(z)$ be corresponding scores of example

$z$. Then we have

$$s_{y_x}^{(x)}(x) = \frac{1}{S}\langle x, \sum_{i=1}^{S} x_i + x - x_S \rangle \tag{A82}$$

$$s_{y_x}^{(z)}(x) = \begin{cases} \frac{1}{S}\langle x, \sum_{i=1}^{S} x_i + z - x_S \rangle & \text{if } y_x = y_z \\ \frac{1}{S}\langle x, \sum_{i=1}^{S} x_i \rangle & \text{if } y_x \neq y_z \end{cases} \tag{A83}$$

$$s_{y_z}^{(x)}(z) = \begin{cases} \frac{1}{S}\langle z, \sum_{i=1}^{S} z_i + x - z_S \rangle & \text{if } y_x = y_z \\ \frac{1}{S}\langle z, \sum_{i=1}^{S} z_i \rangle & \text{if } y_x \neq y_z \end{cases} \tag{A84}$$

$$s_{y_z}^{(z)}(z) = \frac{1}{S}\langle z, \sum_{i=1}^{S} z_i + z - x_S \rangle \tag{A85}$$

where $x_i$ are samples with label $y_x$ and $z_i$ are samples with label $y_z$ when $y_x \neq y_z$. Thus we obtain

$$\mu_{x,x} = \langle x, m_x \rangle + \frac{1}{S}\langle x, x - m_x \rangle \tag{A86}$$

$$\mu_{z,x} = \begin{cases} \langle x, m_x \rangle + \frac{1}{S}\langle x, z - m_x \rangle & \text{if } y_x = y_z \\ \langle x, m_x \rangle & \text{if } y_x \neq y_z \end{cases} \tag{A87}$$

$$\mu_{x,z} = \begin{cases} \langle z, m_x \rangle + \frac{1}{S}\langle z, x - m_x \rangle & \text{if } y_x = y_z \\ \langle z, m_z \rangle & \text{if } y_x \neq y_z \end{cases} \tag{A88}$$

$$\mu_{z,z} = \langle z, m_x \rangle + \frac{1}{S}\langle z, z - m_x \rangle \tag{A89}$$

$$\sigma_x = \frac{1}{\sqrt{S}} s \|x\| \tag{A90}$$

$$\sigma_z = \frac{1}{\sqrt{S}} s \|z\|, \tag{A91}$$

where $m_z$ is the true class mean of $y_z$ when $y_x \neq y_z$ (see Appendix B for derivation of $\sigma_x$ and $\sigma_z$).

Now recall that

$$q = \frac{(\mu_{x,x} - \mu_{z,x})(\hat{\mu}_{x,x} - \hat{\mu}_{z,x})}{\hat{\sigma}_x^2} + \frac{(\mu_{x,z} - \mu_{z,z})(\hat{\mu}_{x,z} - \hat{\mu}_{z,z})}{\hat{\sigma}_z^2} \tag{A92}$$

$$A = \frac{\sigma_x}{\hat{\sigma}_x^2}(\hat{\mu}_{x,x} - \hat{\mu}_{z,x}) + \frac{\sigma_z}{\hat{\sigma}_z^2}(\hat{\mu}_{x,z} - \hat{\mu}_{z,z}) \tag{A93}$$

In the limit of infinitely many shadow models, these can be written as

$$q = \left(\frac{\mu_{x,x} - \mu_{z,x}}{\sigma_x}\right)^2 + \left(\frac{\mu_{x,z} - \mu_{z,z}}{\sigma_z}\right)^2 \tag{A94}$$

$$A = \frac{\mu_{x,x} - \mu_{z,x}}{\sigma_x} + \frac{\mu_{x,z} - \mu_{z,z}}{\sigma_z}. \tag{A95}$$

Using the law of total expectation, we have

$$\mathbb{E}_z[q] = \Pr_z(y_z = y_x)\mathbb{E}_z[q \mid y_z = y_x] + \sum_{j=1, j \neq y_x}^{C} \Pr_z(y_z = j)\mathbb{E}_z[q \mid y_z = j] \tag{A96}$$

$$= \frac{1}{C}\mathbb{E}_z\left[\left(\frac{\langle x, x - z \rangle}{\sqrt{S}s\|x\|}\right)^2 + \left(\frac{\langle z, x - z \rangle}{\sqrt{S}s\|z\|}\right)^2 \,\middle|\, y_z = y_x\right] \tag{A97}$$

$$+ \frac{C-1}{C}\mathbb{E}_z\left[\left(\frac{\langle x, x - m_x \rangle}{\sqrt{S}s\|x\|}\right)^2 + \left(\frac{\langle z, z - m_z \rangle}{\sqrt{S}s\|z\|}\right)^2 \,\middle|\, y_z \neq y_x\right] \tag{A98}$$

$$= \frac{1}{CSs^2}\mathbb{E}_z\left[\frac{\langle x, x - z \rangle^2}{\|x\|^2} + \frac{\langle z, x - z \rangle^2}{\|z\|^2} \,\middle|\, y_z = y_x\right] \tag{A99}$$

$$+ \frac{C-1}{CSs^2}\mathbb{E}_z\left[\frac{\langle x, x - m_x \rangle^2}{\|x\|^2} + \frac{\langle z, z - m_z \rangle^2}{\|z\|^2} \,\middle|\, y_z = y_x\right], \tag{A100}$$

and

$$\mathbb{E}_z[A] = \Pr_z(y_z = y_x)\mathbb{E}_z[A \mid y_z = y_x] + \sum_{j=1, j \neq y_x}^{C} \Pr_z(y_z = j)\mathbb{E}_z[A \mid y_z = j] \tag{A101}$$

$$= \frac{1}{C}\mathbb{E}_z\left[\frac{\langle x, x-z\rangle}{\sqrt{S}s\|x\|} + \frac{\langle z, x-z\rangle}{\sqrt{S}s\|z\|} \;\middle|\; y_z = y_x\right] \tag{A102}$$

$$+ \frac{C-1}{C}\mathbb{E}_z\left[\frac{\langle x, x-m_x\rangle}{\sqrt{S}s\|x\|} + \frac{\langle z, z-m_z\rangle}{\sqrt{S}s\|z\|} \;\middle|\; y_z \neq y_x\right] \tag{A103}$$

$$= \frac{1}{C\sqrt{S}s}\mathbb{E}_z\left[\frac{\langle x, x-z\rangle}{\|x\|} + \frac{\langle z, x-z\rangle}{\|z\|} \;\middle|\; y_z = y_x\right] \tag{A104}$$

$$+ \frac{C-1}{C\sqrt{S}s}\mathbb{E}_z\left[\frac{\langle x, x-m_x\rangle}{\|x\|} + \frac{\langle z, z-m_z\rangle}{\|z\|} \;\middle|\; y_z = y_x\right]. \tag{A105}$$

Hence we obtain by the Cauchy-Schwarz inequality

$$\frac{\mathbb{E}_z[q]}{\mathbb{E}_z[A]} = \frac{1}{\sqrt{S}s} \cdot \frac{\mathbb{E}_z\left[\frac{\langle x, x-z\rangle^2}{\|x\|^2} + \frac{\langle z, x-z\rangle^2}{\|z\|^2} \mid y_z = y_x\right] + (C-1)\mathbb{E}_z\left[\frac{\langle x, x-m_x\rangle^2}{\|x\|^2} + \frac{\langle z, z-m_z\rangle^2}{\|z\|^2} \mid y_z \neq y_x\right]}{\mathbb{E}_z\left[\frac{\langle x, x-z\rangle}{\|x\|} + \frac{\langle z, x-z\rangle}{\|z\|} \mid y_z = y_x\right] + (C-1)\mathbb{E}_z\left[\frac{\langle x, x-m_x\rangle}{\|x\|} + \frac{\langle z, z-m_z\rangle}{\|z\|} \mid y_z \neq y_x\right]} \tag{A106}$$

$$\leq \frac{1}{\sqrt{S}s} \cdot \frac{\mathbb{E}_z\left[2\|x-z\|^2 \mid y_z = y_x\right] + (C-1)\mathbb{E}_z\left[(\|x-m_x\|^2 + \|z-m_z\|^2) \mid y_z \neq y_x\right]}{\mathbb{E}_z\left[2\|x-z\| \mid y_z = y_x\right] + (C-1)\mathbb{E}_z\left[(\|x-m_x\| + \|z-m_z\|) \mid y_z \neq y_x\right]}. \tag{A107}$$

Since the score of the target is normally distributed in the simplified model, by symmetry Lemma 2 yields

$$\mathrm{TPR}_{\mathrm{RMIA}}(x) \leq \Phi\left(\Phi^{-1}(\alpha) + \left|\frac{\mathbb{E}_z[q]}{\mathbb{E}_z[A]}\right|\right). \tag{A108}$$

Thus we have

$$\mathrm{TPR}_{\mathrm{RMIA}}(x) \leq \Phi\left(\Phi^{-1}(\alpha) + \frac{\psi(x, C)}{\sqrt{S}s}\right), \tag{A109}$$

where

$$\psi(x, C) = \frac{\mathbb{E}_z\left[2\|x-z\|^2 \mid y_z = y_x\right] + (C-1)\mathbb{E}_z\left[(\|x-m_x\|^2 + \|z-m_z\|^2) \mid y_z \neq y_x\right]}{\mathbb{E}_z\left[2\|x-z\| \mid y_z = y_x\right] + (C-1)\mathbb{E}_z\left[(\|x-m_x\| + \|z-m_z\|) \mid y_z \neq y_x\right]}. \tag{A110}$$

This completes the first half of the theorem.

Now that

$$\mathrm{TPR}_{\mathrm{RMIA}}(x) = \Pr_\eta\left(\Pr_z(\lambda_z + q \geq \log\gamma) \geq \beta\right) \leq \Pr_\eta\left(\eta \leq \Phi^{-1}(\alpha) + \left|\frac{\mathbb{E}_z[q]}{\mathbb{E}_z[A]}\right|\right) \tag{A111}$$

$$\mathrm{FPR}_{\mathrm{RMIA}}(x) = \Pr_\eta\left(\Pr_z(\lambda_z \geq \log\gamma) \geq \beta\right) \leq \Pr_\eta(\eta \leq \Phi^{-1}(\alpha)) \tag{A112}$$

where $\eta \sim \mathcal{N}(0, 1)$ corresponds to dataset sampling. In the proof of Lemma 2, we derive these upper bound by Markov's inequality and an upper bound of Jensen's gap that is shared for both $\mathrm{TPR}_{\mathrm{RMIA}}(x)$ and $\mathrm{FPR}_{\mathrm{RMIA}}(x)$ cases. Since Markov's inequality is tighter when the threshold is relatively large, inequality (A112) is tighter than inequality (A111). Therefore, for large enough $S$ we obtain

$$\mathrm{TPR}_{\mathrm{RMIA}}(x) - \mathrm{FPR}_{\mathrm{RMIA}}(x) \leq \Pr_\eta\left(\eta \leq \Phi^{-1}(\alpha) + \left|\frac{\mathbb{E}_z[q]}{\mathbb{E}_z[A]}\right|\right) - \Pr_\eta(\eta \leq \Phi^{-1}(\alpha)) \tag{A113}$$

$$\leq \Pr_\eta\left(\eta \leq \Phi^{-1}(\alpha) + \frac{\psi(x, C)}{\sqrt{S}s}\right) - \Pr_\eta(\eta \leq \Phi^{-1}(\alpha)) \tag{A114}$$

$$\approx p_\eta(\Phi^{-1}(\alpha))\frac{\psi(x, C)}{\sqrt{S}} \tag{A115}$$

$$= \frac{1}{\sqrt{2\pi}}e^{-\frac{1}{2}\Phi^{-1}(\alpha)^2}\frac{\psi(x, C)}{\sqrt{S}}. \tag{A116}$$

Hence we have

$$\log(\text{TPR}_{\text{RMIA}}(x) - \text{FPR}_{\text{RMIA}}(x)) \leq -\frac{1}{2}\log S - \frac{1}{2}\Phi^{-1}(\alpha)^2 + \log\frac{\psi(x,C)}{\sqrt{2\pi}}. \tag{A117}$$

$\square$

### C.5 PROOF OF COROLLARY 5

**Corollary 5** (Average-case LiRA power-law). *For the simplified model with arbitrary $C$, sufficiently large $S$ and infinitely many shadow models, we have*

$$\log(\overline{\text{TPR}}_{\text{LiRA}} - \overline{\text{FPR}}_{\text{LiRA}}) \approx -\frac{1}{2}\log S - \frac{1}{2}\Phi^{-1}(\overline{\text{FPR}}_{\text{LiRA}})^2 + \log\left(\mathbb{E}_{(x,y_x)\sim\mathbb{D}}\left[\frac{\langle x, x - m_x\rangle}{\sqrt{2\pi}||x||s}\right]\right). \tag{24}$$

*Proof.* By theorem 3 and the law of unconscious statistician, we have for large $S$

$$\overline{\text{TPR}}_{\text{LiRA}} - \overline{\text{FPR}}_{\text{LiRA}} = \int_{\mathbb{D}} \text{Pr}(x)(\text{TPR}_{\text{LiRA}}(x) - \text{FPR}_{\text{LiRA}}(x))dx \tag{A118}$$

$$\approx \int_{\mathbb{D}} \text{Pr}(x)\frac{1}{\sqrt{2\pi}}e^{-\frac{1}{2}\Phi^{-1}(\overline{\text{FPR}}_{\text{LiRA}})^2}\frac{\langle x, x - m_x\rangle}{\sqrt{S}||x||s}dx \tag{A119}$$

$$= \frac{1}{\sqrt{2\pi}}e^{-\frac{1}{2}\Phi^{-1}(\overline{\text{FPR}}_{\text{LiRA}})^2}\frac{1}{\sqrt{S}}\int_{\mathbb{D}} \text{Pr}(x)\frac{\langle x, x - m_x\rangle}{||x||s}dx \tag{A120}$$

$$= \frac{1}{\sqrt{S}}e^{-\frac{1}{2}\Phi^{-1}(\overline{\text{FPR}}_{\text{LiRA}})^2}\mathbb{E}_{(x,y_x)\sim\mathbb{D}}\left[\frac{\langle x, x - m_x\rangle}{\sqrt{2\pi}||x||s}\right]. \tag{A121}$$

Note that here we fixed $\text{FPR}_{\text{LiRA}}(x) = \overline{\text{FPR}}_{\text{LiRA}}$ for all $x$. Then we obtain

$$\log(\overline{\text{TPR}}_{\text{LiRA}} - \overline{\text{FPR}}_{\text{LiRA}}) \approx -\frac{1}{2}\log S - \frac{1}{2}\Phi^{-1}(\text{FPR})^2 + \log\left(\mathbb{E}_{(x,y_x)\sim\mathbb{D}}\left[\frac{\langle x, x - m_x\rangle}{\sqrt{2\pi}||x||s}\right]\right). \tag{A122}$$

$\square$

### C.6 PROOF OF COROLLARY 6

**Corollary 6** (Average-case RMIA power-law). *For the simplified model with sufficiently large $S$ and infinitely many shadow models, we have*

$$\log(\overline{\text{TPR}}_{\text{RMIA}} - \overline{\text{FPR}}_{\text{RMIA}}) \leq -\frac{1}{2}\log S - \frac{1}{2}\Phi^{-1}(\alpha)^2 + \log\left(\mathbb{E}_{(x,y_x)\sim\mathbb{D}}\left[\frac{\psi(x,C)}{\sqrt{2\pi}}\right]\right). \tag{25}$$

*Proof.* By theorem 4 and the law of unconscious statistician, we have for large $S$

$$\overline{\text{TPR}}_{\text{RMIA}} - \overline{\text{FPR}}_{\text{RMIA}} = \int_{\mathbb{D}} \text{Pr}(x)(\text{TPR}_{\text{RMIA}}(x) - \text{FPR}_{\text{RMIA}}(x))dx \tag{A123}$$

$$\leq \int_{\mathbb{D}} \text{Pr}(x)\frac{1}{\sqrt{2\pi}}e^{-\frac{1}{2}\Phi^{-1}(\alpha)^2}\frac{\psi(x,C)}{\sqrt{S}}dx \tag{A124}$$

$$= \frac{1}{\sqrt{2\pi}}e^{-\frac{1}{2}\Phi^{-1}(\alpha)^2}\int_{\mathbb{D}}\frac{\psi(x,C)}{\sqrt{S}}dx. \tag{A125}$$

$$= \frac{1}{\sqrt{S}}e^{-\frac{1}{2}\Phi^{-1}(\alpha)^2}\mathbb{E}_{(x,y_x)\sim\mathbb{D}}\left[\frac{\psi(x,C)}{\sqrt{2\pi}}\right]. \tag{A126}$$

Hence we obtain

$$\log(\overline{\text{TPR}}_{\text{RMIA}} - \overline{\text{FPR}}_{\text{RMIA}}) \leq -\frac{1}{2}\log S - \frac{1}{2}\Phi^{-1}(\alpha)^2 + \log\left(\mathbb{E}_{(x,y_x)\sim\mathbb{D}}\left[\frac{\psi(x,C)}{\sqrt{2\pi}}\right]\right). \tag{A127}$$

$\square$

## D    LiRA VULNERABILITY FOR SMALL FPR

In Section 3.4 we proved for the simplified model that for LiRA $\log(\text{TPR} - \text{FPR}) \approx -\frac{1}{2}\log S$ ignoring additive constants when FPR is fixed (Corollary 5). In Section 4.3 we observed that the coefficient $\beta_S$ for $\log S$ is around $-0.5$ for larger FPR, aligning with the theoretical value. However, for smaller FPR the coefficient is smaller than $-0.5$. To understand this phenomenon, it is important to note that the power-law (Theorem 3 and Corollary 5) only holds for sufficiently large $S$. Thus, it can be explained that the difference of coefficient values $\beta_S$ for small and large FPR comes from the small-$S$ regime as follows.

In the proof of Corollary 5 and Theorem 3 the only approximation that could introduce some bias is Equation (A80). That is,

$$\text{TPR}_{\text{LiRA}}(x) - \text{FPR}_{\text{LiRA}}(x) = \Pr\left(\eta \leq \Phi^{-1}(\text{FPR}_{\text{LiRA}}(x)) + r\right) - \Pr\left(\eta \leq \Phi^{-1}(\text{FPR}_{\text{LiRA}}(x))\right) \tag{A128}$$

$$\approx p_\eta\left(\Phi^{-1}(\text{FPR}_{\text{LiRA}}(x))\right) r, \tag{A129}$$

where $\eta \sim \mathcal{N}(0,1)$, $r$ is a shift that scales $O(1/\sqrt{S})$ and $p_\eta$ is the pdf of $\eta$. Figure A.1 numerically illustrates this approximation. It can be observed that for small $S$ the approximation does not hold, underestimating the true vulnerability. Particularly, this effect is remarkably larger for small $\text{FPR}_{\text{LiRA}}(x)$. In other words, the simplified model overestimates the coefficient value $\beta_S$ for small $\text{FPR}_{\text{LiRA}}(x)$ by underestimating the true vulnerability in the small-$S$ regime.

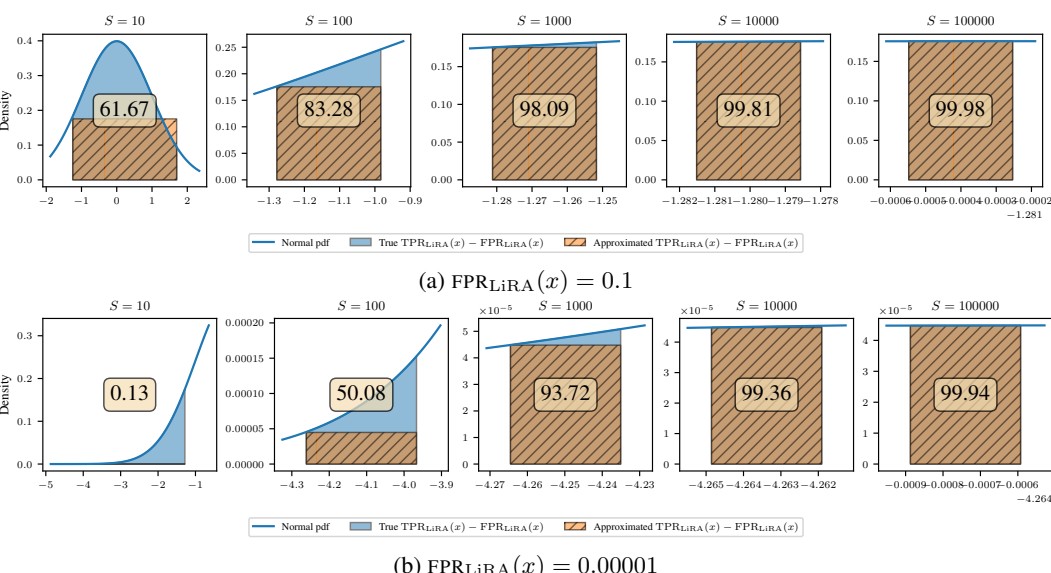

(a) $\text{FPR}_{\text{LiRA}}(x) = 0.1$

(b) $\text{FPR}_{\text{LiRA}}(x) = 0.00001$

Figure A.1: True vs. approximated $\text{TPR}_{\text{LiRA}}(x) - \text{FPR}_{\text{LiRA}}(x)$ for the simplified model with in-class standard deviation $s = 0.1$ and dimension $d = 100$. One target example $(x, y_x)$ is randomly sampled from the underlying distribution. Numbers in the plots are percentages of the approximated vulnerabilities over the true vulnerabilities. Figures illustrate the approximation for different $S$ (a) when $\text{FPR}_{\text{LiRA}}(x) = 0.1$ and (b) when $\text{FPR}_{\text{LiRA}}(x) = 0.00001$.

# E  TRAINING DETAILS

## E.1  PARAMETERIZATION

We utilise pre-trained feature extractors BiT-M-R50x1 (R-50) (Kolesnikov et al., 2020) with 23.5M parameters and Vision Transformer ViT-Base-16 (ViT-B) (Dosovitskiy et al., 2021) with 85.8M parameters, both pretrained on the ImageNet-21K dataset (Russakovsky et al., 2015). We download the feature extractor checkpoints from the respective repositories.

Following Tobaben et al. (2023) that show the favorable trade-off of parameter-efficient fine-tuning between computational cost, utility and privacy even for small datasets, we only consider fine-tuning subsets of all feature extractor parameters. We consider the following configurations:

- **Head:** We train a linear layer on top of the pre-trained feature extractor.
- **FiLM:** In addition to the linear layer from Head, we fine-tune parameter-efficient FiLM (Perez et al., 2018) adapters scattered throughout the network. While a diverse set of adapters has been proposed, we utilise FiLM as it has been shown to be competitive in prior work (Shysheya et al., 2023; Tobaben et al., 2023).

### E.1.1  LICENSES AND ACCESS

The licenses and means to access the model checkpoints can be found below.

- BiT-M-R50x1 (R-50) (Kolesnikov et al., 2020) is licensed with the Apache-2.0 license and can be obtained through the instructions on `https://github.com/google-research/big_transfer`.
- Vision Transformer ViT-Base-16 (ViT-B) (Dosovitskiy et al., 2021) is licensed with the Apache-2.0 license and can be obtained through the instructions on `https://github.com/google-research/vision_transformer`.

## E.2  HYPERPARAMETER TUNING

Our hyperparameter tuning is heavily inspired by the comprehensive few-shot experiments by Tobaben et al. (2023). We utilise their hyperparameter tuning protocol as it has been proven to yield SOTA results for (DP) few-shot models. Given the input $\mathcal{D}$ dataset we perform hyperparameter tuning by splitting the $\mathcal{D}$ into 70% train and 30% validation. We then perform the specified iterations of hyperparameter tuning using the tree-structured Parzen estimator (Bergstra et al., 2011) strategy as implemented in Optuna (Akiba et al., 2019) to derive a set of hyperparameters that yield the highest accuracy on the validation split. This set of hyperparameters is subsequently used to train all shadow models with the Adam optimizer (Kingma & Ba, 2015). Details on the set of hyperparameters that are tuned and their ranges can be found in Table A1.

Table A1: Hyperparameter ranges used for the Bayesian optimization with Optuna.

|  | lower bound | upper bound |
| --- | --- | --- |
| batch size | 10 | $|\mathcal{D}|$ |
| clipping norm | 0.2 | 10 |
| epochs | 1 | 200 |
| learning rate | 1e-7 | 1e-2 |

## E.3  DATASETS

Table A2 shows the datasets used in the paper. We base our experiments on a subset of the the few-shot benchmark VTAB (Zhai et al., 2019) that achieves a classification accuracy $> 80\%$ and thus would considered to be used by a practitioner. Additionally, we add CIFAR10 which is not part of the original VTAB benchmark.

Table A2: Used datasets in the paper, their minimum and maximum shots $S$ and maximum number of classes $C$ and their test accuracy when fine-tuning a non-DP ViT-B Head. The test accuracy for EuroSAT and Resics45 is computed on the part of the training split that is not used for training the particular model due to both datasets missing an official test split. Note that LiRA requires $2S$ for training the shadow models and thus $S$ is smaller than when only performing fine-tuning.

| dataset | (max.) $C$ | min. $S$ | max. $S$ | test accuracy (min. $S$) | test accuracy (max. $S$) |
|---|---|---|---|---|---|
| Patch Camelyon (Veeling et al., 2018) | 2 | 256 | 65536 | 82.8% | 85.6% |
| CIFAR10 (Krizhevsky, 2009) | 10 | 8 | 2048 | 92.7% | 97.7% |
| EuroSAT (Helber et al., 2019) | 10 | 8 | 512 | 80.2% | 96.7% |
| Pets (Parkhi et al., 2012) | 37 | 8 | 32 | 82.3% | 90.7% |
| Resics45 (Cheng et al., 2017) | 45 | 32 | 256 | 83.5% | 91.6% |
| CIFAR100 (Krizhevsky, 2009) | 100 | 16 | 128 | 82.2% | 87.6% |

### E.3.1 LICENSES AND ACCESS

The licenses and means to access the datasets can be found below. We downloaded all datasets from TensorFlow datasets https://www.tensorflow.org/datasets but Resics45 which required manual download.

- Patch Camelyon (Veeling et al., 2018) is licensed with Creative Commons Zero v1.0 Universal (cc0-1.0) and we use version 2.0.0 of the dataset as specified on https://www.tensorflow.org/datasets/catalog/patch_camelyon.

- CIFAR10 (Krizhevsky, 2009) is licensed with an unknown license and we use version 3.0.2 of the dataset as specified on https://www.tensorflow.org/datasets/catalog/cifar10.

- EuroSAT (Helber et al., 2019) is licensed with MIT and we use version 2.0.0 of the dataset as specified on https://www.tensorflow.org/datasets/catalog/eurosat.

- Pets (Parkhi et al., 2012) is licensed with CC BY-SA 4.0 Deed and we use version 3.2.0 of the dataset as specified on https://www.tensorflow.org/datasets/catalog/oxford_iiit_pet.

- Resics45 (Cheng et al., 2017) is licensed with an unknown license and we use version 3.0.0 of the dataset as specified on https://www.tensorflow.org/datasets/catalog/resisc45.

- CIFAR100 (Krizhevsky, 2009) is licensed with an unknown license and we use version 3.0.2 of the dataset as specified on https://www.tensorflow.org/datasets/catalog/cifar100.

### E.4 COMPUTE RESOURCES

All experiments but the R-50 (FiLM) experiments are run on CPU with 8 cores and 16 GB of host memory. The training time depends on the model (ViT is cheaper than R-50), number of shots $S$ and the number of classes $C$ but ranges for the training of one model from some minutes to an hour. This assumes that the images are passed once through the pre-trained backbone and then cached as feature vectors. The provided code implements this optimization.

The R-50 (FiLM) experiments are significantly more expensive and utilise a NVIDIA V100 with 40 GB VRAM, 10 CPU cores and 64 GB of host memory. The training of 257 shadow models then does not exceed 24h for the settings that we consider.

We estimate that in total we spend around 7 days of V100 and some dozens of weeks of CPU core time but more exact measurements are hard to make.

# F   ADDITIONAL RESULTS

In this section, we provide tabular results for our experiments and additional figures that did not fit into the main paper.

## F.1   ADDITIONAL RESULTS FOR SECTION 4

This Section contains additional results for Section 4.

### F.1.1   VULNERABILITY AS A FUNCTION OF SHOTS

This section displays additional results to Figure 1a for FPR $\in \{0.1, 0.01, 0.001\}$ for ViT-B and R-50 in in Figure A.2 and Tables A3 and A4.

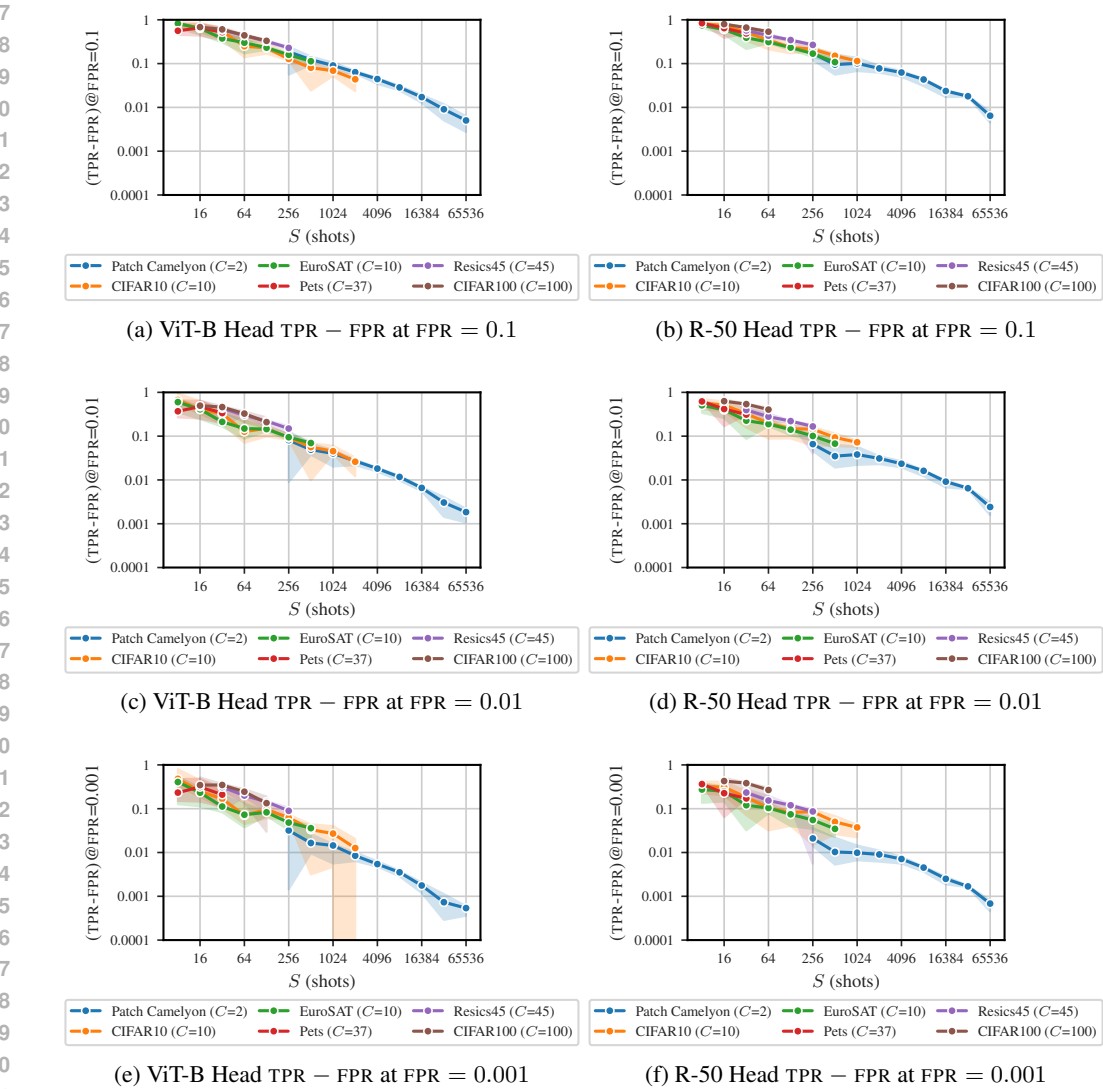

(a) ViT-B Head TPR − FPR at FPR = 0.1     (b) R-50 Head TPR − FPR at FPR = 0.1

(c) ViT-B Head TPR − FPR at FPR = 0.01    (d) R-50 Head TPR − FPR at FPR = 0.01

(e) ViT-B Head TPR − FPR at FPR = 0.001   (f) R-50 Head TPR − FPR at FPR = 0.001

Figure A.2: MIA vulnerability as a function of shots (examples per class) when attacking a pre-trained ViT-B and R-50 Head trained without DP on different downstream datasets. The errorbars display the minimum and maximum Clopper-Pearson CIs over six seeds and the solid line the median.

Table A3: Median MIA vulnerability over six seeds as a function of $S$ (shots) when attacking a Head trained without DP on-top of a ViT-B. The ViT-B is pre-trained on ImageNet-21k.

| dataset | classes ($C$) | shots ($S$) | tpr@fpr=0.1 | tpr@fpr=0.01 | tpr@fpr=0.001 |
|---|---|---|---|---|---|
| Patch Camelyon (Veeling et al., 2018) | 2 | 256 | 0.266 | 0.086 | 0.032 |
| | | 512 | 0.223 | 0.059 | 0.018 |
| | | 1024 | 0.191 | 0.050 | 0.015 |
| | | 2048 | 0.164 | 0.037 | 0.009 |
| | | 4096 | 0.144 | 0.028 | 0.007 |
| | | 8192 | 0.128 | 0.021 | 0.005 |
| | | 16384 | 0.118 | 0.017 | 0.003 |
| | | 32768 | 0.109 | 0.014 | 0.002 |
| | | 65536 | 0.105 | 0.012 | 0.002 |
| CIFAR10 (Krizhevsky, 2009) | 10 | 8 | 0.910 | 0.660 | 0.460 |
| | | 16 | 0.717 | 0.367 | 0.201 |
| | | 32 | 0.619 | 0.306 | 0.137 |
| | | 64 | 0.345 | 0.132 | 0.067 |
| | | 128 | 0.322 | 0.151 | 0.082 |
| | | 256 | 0.227 | 0.096 | 0.054 |
| | | 512 | 0.190 | 0.068 | 0.032 |
| | | 1024 | 0.168 | 0.056 | 0.025 |
| | | 2048 | 0.148 | 0.039 | 0.013 |
| EuroSAT (Helber et al., 2019) | 10 | 8 | 0.921 | 0.609 | 0.408 |
| | | 16 | 0.738 | 0.420 | 0.234 |
| | | 32 | 0.475 | 0.222 | 0.113 |
| | | 64 | 0.400 | 0.159 | 0.074 |
| | | 128 | 0.331 | 0.155 | 0.084 |
| | | 256 | 0.259 | 0.104 | 0.049 |
| | | 512 | 0.213 | 0.080 | 0.037 |
| Pets (Parkhi et al., 2012) | 37 | 8 | 0.648 | 0.343 | 0.160 |
| | | 16 | 0.745 | 0.439 | 0.259 |
| | | 32 | 0.599 | 0.311 | 0.150 |
| Resics45 (Cheng et al., 2017) | 45 | 32 | 0.672 | 0.425 | 0.267 |
| | | 64 | 0.531 | 0.295 | 0.168 |
| | | 128 | 0.419 | 0.212 | 0.115 |
| | | 256 | 0.323 | 0.146 | 0.072 |
| CIFAR100 (Krizhevsky, 2009) | 100 | 16 | 0.814 | 0.508 | 0.324 |
| | | 32 | 0.683 | 0.445 | 0.290 |
| | | 64 | 0.538 | 0.302 | 0.193 |
| | | 128 | 0.433 | 0.208 | 0.114 |

Table A4: Median MIA vulnerability over six seeds as a function of $S$ (shots) when attacking a Head trained without DP on-top of a R-50. The R-50 is pre-trained on ImageNet-21k.

| dataset | classes ($C$) | shots ($S$) | tpr@fpr=0.1 | tpr@fpr=0.01 | tpr@fpr=0.001 |
|---|---|---|---|---|---|
| Patch Camelyon (Veeling et al., 2018) | 2 | 256 | 0.272 | 0.076 | 0.022 |
| | | 512 | 0.195 | 0.045 | 0.011 |
| | | 1024 | 0.201 | 0.048 | 0.011 |
| | | 2048 | 0.178 | 0.041 | 0.010 |
| | | 4096 | 0.163 | 0.033 | 0.008 |
| | | 8192 | 0.143 | 0.026 | 0.006 |
| | | 16384 | 0.124 | 0.019 | 0.004 |
| | | 32768 | 0.118 | 0.016 | 0.003 |
| | | 65536 | 0.106 | 0.012 | 0.002 |
| CIFAR10 (Krizhevsky, 2009) | 10 | 8 | 0.911 | 0.574 | 0.324 |
| | | 16 | 0.844 | 0.526 | 0.312 |
| | | 32 | 0.617 | 0.334 | 0.183 |
| | | 64 | 0.444 | 0.208 | 0.106 |
| | | 128 | 0.334 | 0.159 | 0.084 |
| | | 256 | 0.313 | 0.154 | 0.086 |
| | | 512 | 0.251 | 0.103 | 0.051 |
| | | 1024 | 0.214 | 0.082 | 0.038 |
| EuroSAT (Helber et al., 2019) | 10 | 8 | 0.846 | 0.517 | 0.275 |
| | | 16 | 0.699 | 0.408 | 0.250 |
| | | 32 | 0.490 | 0.236 | 0.121 |
| | | 64 | 0.410 | 0.198 | 0.105 |
| | | 128 | 0.332 | 0.151 | 0.075 |
| | | 256 | 0.269 | 0.111 | 0.056 |
| | | 512 | 0.208 | 0.077 | 0.036 |
| Pets (Parkhi et al., 2012) | 37 | 8 | 0.937 | 0.631 | 0.366 |
| | | 16 | 0.745 | 0.427 | 0.227 |
| | | 32 | 0.588 | 0.321 | 0.173 |
| Resics45 (Cheng et al., 2017) | 45 | 32 | 0.671 | 0.405 | 0.235 |
| | | 64 | 0.534 | 0.289 | 0.155 |
| | | 128 | 0.445 | 0.231 | 0.121 |
| | | 256 | 0.367 | 0.177 | 0.088 |
| CIFAR100 (Krizhevsky, 2009) | 100 | 16 | 0.897 | 0.638 | 0.429 |
| | | 32 | 0.763 | 0.549 | 0.384 |
| | | 64 | 0.634 | 0.414 | 0.269 |

## F.1.2 VULNERABILITY AS A FUNCTION OF THE NUMBER OF CLASSES

This section displays additional results to Figure 1b for FPR $\in \{0.1, 0.01, 0.001\}$ for ViT-B and R-50 in in Figure A.3 and Tables A5 and A6.

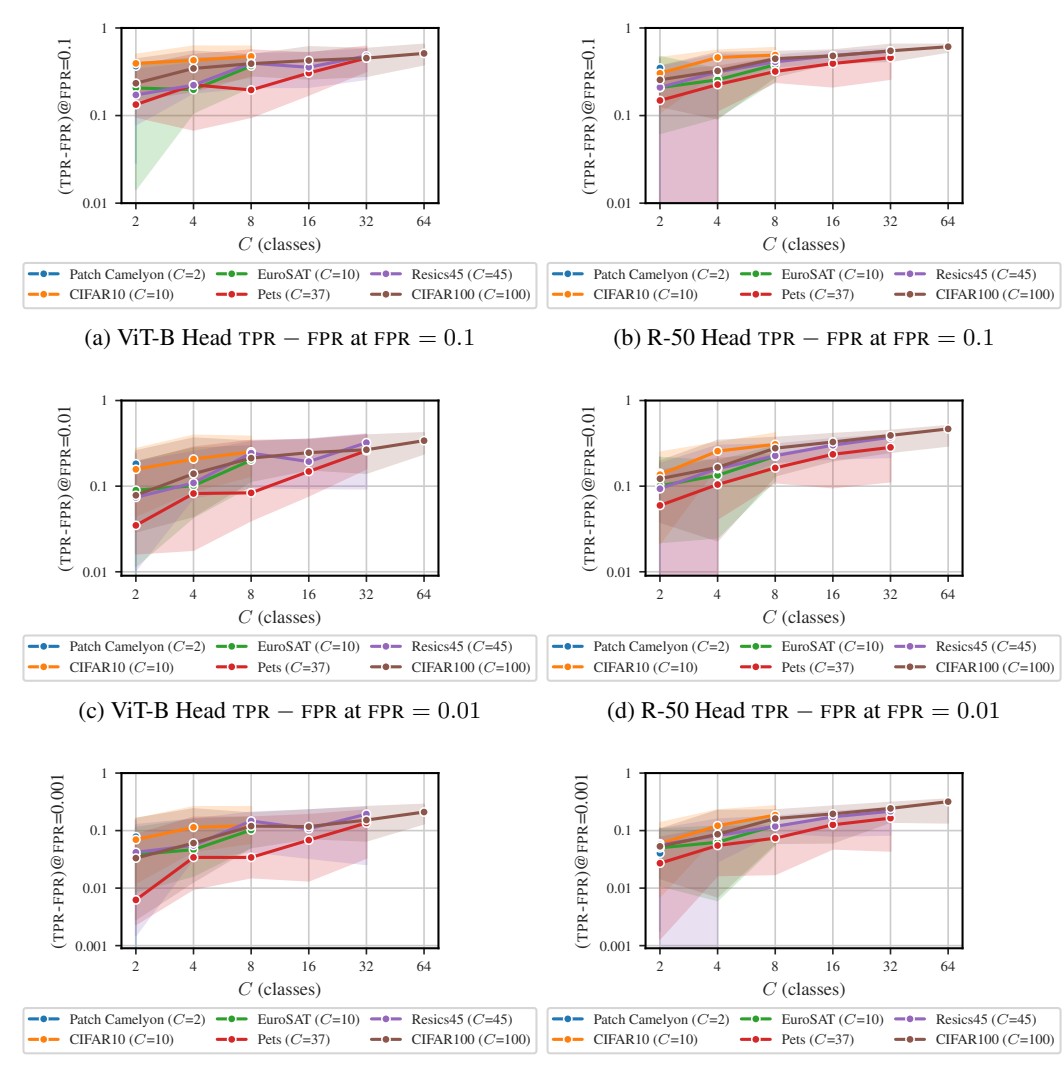

(a) ViT-B Head TPR − FPR at FPR = 0.1

(b) R-50 Head TPR − FPR at FPR = 0.1

(c) ViT-B Head TPR − FPR at FPR = 0.01

(d) R-50 Head TPR − FPR at FPR = 0.01

(e) ViT-B Head TPR − FPR at FPR = 0.001

(f) R-50 Head TPR − FPR at FPR = 0.001

Figure A.3: MIA vulnerability as a function of $C$ (classes) when attacking a ViT-B and R-50 Head fine-tuned without DP on different datasets where the classes are randomly sub-sampled and $S = 32$. The solid line displays the median and the errorbars the min and max clopper-pearson CIs over 12 seeds.

Table A5: Median MIA vulnerability over 12 seeds as a function of $C$ (classes) when attacking a Head trained without DP on-top of a ViT-B. The Vit-B is pre-trained on ImageNet-21k.

| dataset | shots ($S$) | classes ($C$) | tpr@fpr=0.1 | tpr@fpr=0.01 | tpr@fpr=0.001 |
|---|---|---|---|---|---|
| Patch Camelyon (Veeling et al., 2018) | 32 | 2 | 0.467 | 0.192 | 0.080 |
| CIFAR10 (Krizhevsky, 2009) | 32 | 2 | 0.494 | 0.167 | 0.071 |
| | | 4 | 0.527 | 0.217 | 0.115 |
| | | 8 | 0.574 | 0.262 | 0.123 |
| EuroSAT (Helber et al., 2019) | 32 | 2 | 0.306 | 0.100 | 0.039 |
| | | 4 | 0.298 | 0.111 | 0.047 |
| | | 8 | 0.468 | 0.211 | 0.103 |
| Pets (Parkhi et al., 2012) | 32 | 2 | 0.232 | 0.045 | 0.007 |
| | | 4 | 0.324 | 0.092 | 0.035 |
| | | 8 | 0.296 | 0.094 | 0.035 |
| | | 16 | 0.406 | 0.158 | 0.069 |
| | | 32 | 0.553 | 0.269 | 0.136 |
| Resics45 (Cheng et al., 2017) | 32 | 2 | 0.272 | 0.084 | 0.043 |
| | | 4 | 0.322 | 0.119 | 0.056 |
| | | 8 | 0.496 | 0.253 | 0.148 |
| | | 16 | 0.456 | 0.204 | 0.108 |
| | | 32 | 0.580 | 0.332 | 0.195 |
| CIFAR100 (Krizhevsky, 2009) | 32 | 2 | 0.334 | 0.088 | 0.035 |
| | | 4 | 0.445 | 0.150 | 0.061 |
| | | 8 | 0.491 | 0.223 | 0.121 |
| | | 16 | 0.525 | 0.256 | 0.118 |
| | | 32 | 0.553 | 0.276 | 0.153 |
| | | 64 | 0.612 | 0.350 | 0.211 |

Table A6: Median MIA vulnerability over 12 seeds as a function of $C$ (classes) when attacking a Head trained without DP on-top of a R-50. The R-50 is pre-trained on ImageNet-21k.

| dataset | shots ($S$) | classes ($C$) | tpr@fpr=0.1 | tpr@fpr=0.01 | tpr@fpr=0.001 |
|---|---|---|---|---|---|
| Patch Camelyon (Veeling et al., 2018) | 32 | 2 | 0.452 | 0.151 | 0.041 |
| CIFAR10 (Krizhevsky, 2009) | 32 | 2 | 0.404 | 0.146 | 0.060 |
| | | 4 | 0.560 | 0.266 | 0.123 |
| | | 8 | 0.591 | 0.318 | 0.187 |
| EuroSAT (Helber et al., 2019) | 32 | 2 | 0.309 | 0.111 | 0.050 |
| | | 4 | 0.356 | 0.144 | 0.064 |
| | | 8 | 0.480 | 0.233 | 0.123 |
| Pets (Parkhi et al., 2012) | 32 | 2 | 0.249 | 0.068 | 0.029 |
| | | 4 | 0.326 | 0.115 | 0.056 |
| | | 8 | 0.419 | 0.173 | 0.075 |
| | | 16 | 0.493 | 0.245 | 0.127 |
| | | 32 | 0.559 | 0.294 | 0.166 |
| Resics45 (Cheng et al., 2017) | 32 | 2 | 0.310 | 0.103 | 0.059 |
| | | 4 | 0.415 | 0.170 | 0.083 |
| | | 8 | 0.510 | 0.236 | 0.119 |
| | | 16 | 0.585 | 0.311 | 0.174 |
| | | 32 | 0.644 | 0.382 | 0.218 |
| CIFAR100 (Krizhevsky, 2009) | 32 | 2 | 0.356 | 0.132 | 0.054 |
| | | 4 | 0.423 | 0.176 | 0.087 |
| | | 8 | 0.545 | 0.288 | 0.163 |
| | | 16 | 0.580 | 0.338 | 0.196 |
| | | 32 | 0.648 | 0.402 | 0.244 |
| | | 64 | 0.711 | 0.476 | 0.320 |

### F.1.3  DATA FOR FiLM AND FROM SCRATCH TRAINING

Table A7: MIA vulnerability data used in Figure 4b. Note that the data from Carlini et al. (2022) is only partially tabular, thus we estimated the TPR at FPR from the plots in the Appendix of their paper.

| model | dataset | classes (C) | shots (S) | source | tpr@ fpr=0.1 | tpr@ fpr=0.01 | tpr@ fpr=0.001 |
|---|---|---|---|---|---|---|---|
| R-50 FiLM | CIFAR10 (Krizhevsky, 2009) | 10 | 50 | This work | 0.482 | 0.275 | 0.165 |
| | CIFAR100 (Krizhevsky, 2009) | 100 | 10 | Tobaben et al. (2023) | 0.933 | 0.788 | 0.525 |
| | | | 25 | Tobaben et al. (2023) | 0.766 | 0.576 | 0.449 |
| | | | 50 | Tobaben et al. (2023) | 0.586 | 0.388 | 0.227 |
| | | | 100 | Tobaben et al. (2023) | 0.448 | 0.202 | 0.077 |
| | EuroSAT (Helber et al., 2019) | 10 | 8 | This work | 0.791 | 0.388 | 0.144 |
| | Patch Camelyon (Veeling et al., 2018) | 2 | 256 | This work | 0.379 | 0.164 | 0.076 |
| | Pets (Parkhi et al., 2012) | 37 | 8 | This work | 0.956 | 0.665 | 0.378 |
| | Resics45 (Cheng et al., 2017) | 45 | 32 | This work | 0.632 | 0.379 | 0.217 |
| from scratch | CIFAR10 (Krizhevsky, 2009) | 10 | 2500 | Carlini et al. (2022) | 0.300 | 0.110 | 0.084 |
| (wide ResNet) | CIFAR100 (Krizhevsky, 2009) | 100 | 250 | Carlini et al. (2022) | 0.700 | 0.400 | 0.276 |

### F.1.4  PREDICTING DATASET VULNERABILITY AS FUNCTION OF $S$ AND $C$

This section provides additional results for the model based on Equation (27)

Table A8: Results for fitting Equation (A130) with statsmodels Seabold & Perktold (2010) to ViT Head data at FPR $\in \{0.1, 0.01, 0.001, 0.0001, 0.00001\}$. We utilize an ordinary least squares. The test $R^2$ assesses the fit to the data of R-50 Head.

| coeff. | FPR | $R^2$ | test $R^2$ | coeff. value | std. error | $t$ | $p > |z|$ | coeff. [0.025 | coeff. 0.975] |
|---|---|---|---|---|---|---|---|---|---|
| $\beta_S$ (for $S$) | 0.1 | 0.952 | 0.907 | -0.506 | 0.011 | -44.936 | 0.000 | -0.529 | -0.484 |
| | 0.01 | 0.946 | 0.854 | -0.555 | 0.014 | -39.788 | 0.000 | -0.582 | -0.527 |
| | 0.001 | 0.930 | 0.790 | -0.627 | 0.019 | -32.722 | 0.000 | -0.664 | -0.589 |
| | 0.0001 | 0.852 | 0.618 | -0.741 | 0.035 | -21.467 | 0.000 | -0.809 | -0.673 |
| | 0.00001 | 0.837 | 0.404 | -0.836 | 0.045 | -18.690 | 0.000 | -0.924 | -0.748 |
| $\beta_C$ (for $C$) | 0.1 | 0.952 | 0.907 | 0.090 | 0.021 | 4.231 | 0.000 | 0.048 | 0.131 |
| | 0.01 | 0.946 | 0.854 | 0.182 | 0.026 | 6.960 | 0.000 | 0.131 | 0.234 |
| | 0.001 | 0.930 | 0.790 | 0.300 | 0.036 | 8.335 | 0.000 | 0.229 | 0.371 |
| | 0.0001 | 0.852 | 0.618 | 0.363 | 0.065 | 5.616 | 0.000 | 0.236 | 0.491 |
| | 0.00001 | 0.837 | 0.404 | 0.569 | 0.085 | 6.655 | 0.000 | 0.400 | 0.737 |
| $\beta_0$ (intercept) | 0.1 | 0.952 | 0.907 | 0.314 | 0.045 | 6.953 | 0.000 | 0.225 | 0.402 |
| | 0.01 | 0.946 | 0.854 | 0.083 | 0.056 | 1.491 | 0.137 | -0.027 | 0.193 |
| | 0.001 | 0.930 | 0.790 | -0.173 | 0.077 | -2.261 | 0.025 | -0.324 | -0.022 |
| | 0.0001 | 0.852 | 0.618 | -0.303 | 0.138 | -2.202 | 0.029 | -0.575 | -0.032 |
| | 0.00001 | 0.837 | 0.404 | -0.615 | 0.180 | -3.414 | 0.001 | -0.970 | -0.260 |

Figure A.4 shows the performance for all considered FPR.

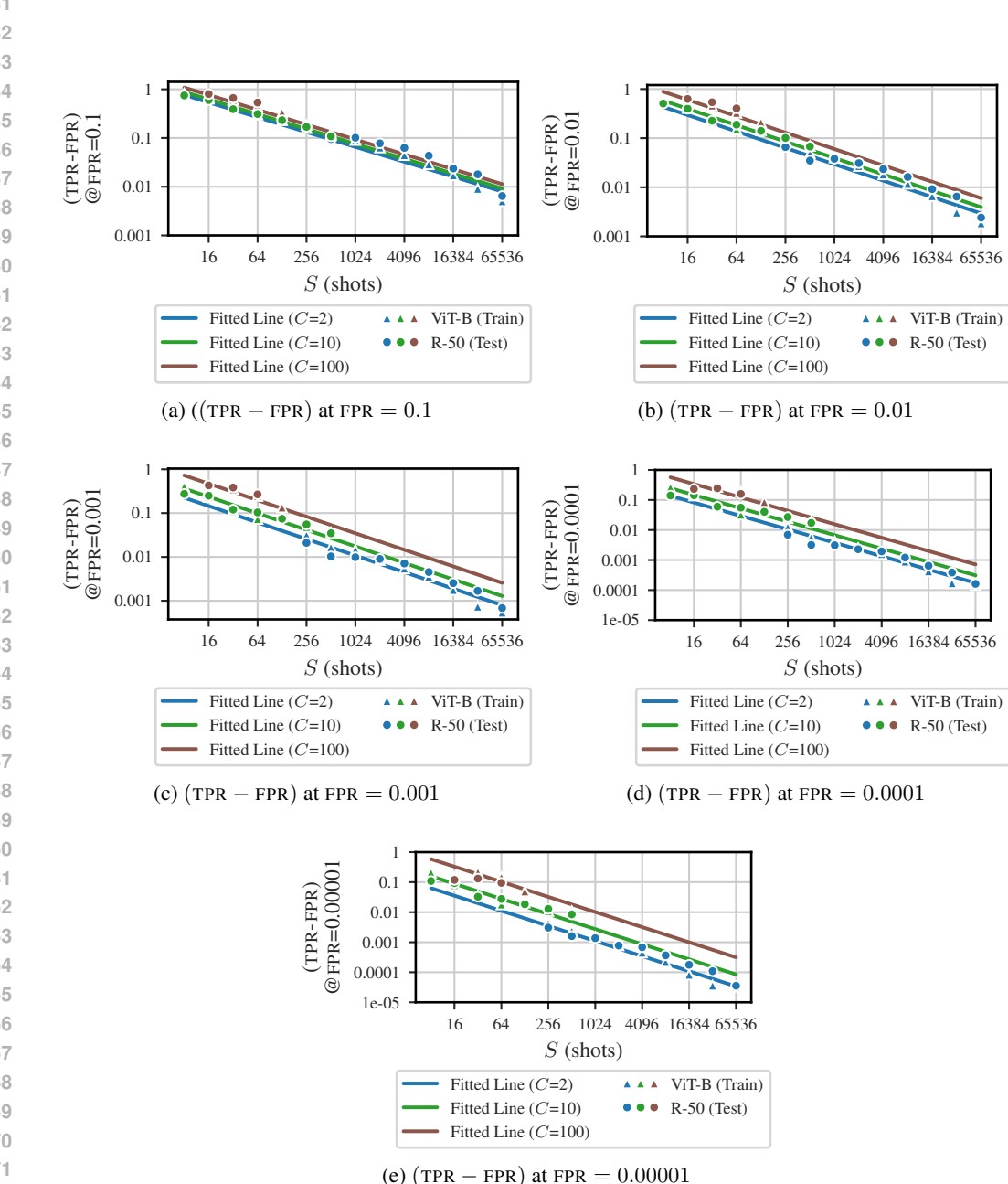

(a) $(\text{TPR} - \text{FPR})$ at $\text{FPR} = 0.1$

(b) $(\text{TPR} - \text{FPR})$ at $\text{FPR} = 0.01$

(c) $(\text{TPR} - \text{FPR})$ at $\text{FPR} = 0.001$

(d) $(\text{TPR} - \text{FPR})$ at $\text{FPR} = 0.0001$

(e) $(\text{TPR} - \text{FPR})$ at $\text{FPR} = 0.00001$

Figure A.4: Predicted MIA vulnerability as a function of $S$ (shots) using a model based on Equation (27) fitted Table A3 (ViT-B). The triangles show the median $\text{TPR} - \text{FPR}$ for the train set (ViT-B; Table A3) and circle the test set (R-50; Table A4) over six seeds. Note that the triangles and dots for $C = 10$ are for EuroSAT.

## F.2 SIMPLER VARIANT OF THE PREDICTION MODEL

The prediction model in the main text (Equation (27)) avoids predicting TPR < FPR in the tail when $S$ is very large. In this section, we analyse a variation of the regression model that is simpler and predicts $\log_{10}(\text{TPR})$ instead of $\log_{10}(\text{TPR} - \text{FPR})$. This variation fits worse to the empirical data and will predict TPR < FPR for high $S$.

The general form this variant can be found in Equation (A130), where $\beta_S, \beta_C$ and $\beta_0$ are the learnable regression parameters.

$$\log_{10}(\text{TPR}) = \beta_S \log_{10}(S) + \beta_C \log_{10}(C) + \beta_0 \tag{A130}$$

Table A9 provides tabular results on the performance of the variant.

Table A9: Results for fitting Equation (A130) with statsmodels Seabold & Perktold (2010) to ViT Head data at FPR $\in \{0.1, 0.01, 0.001, 0.0001, 0.00001\}$. We utilize an ordinary least squares. The test $R^2$ assesses the fit to the data of R-50 Head.

| coeff. | FPR | $R^2$ | test $R^2$ | coeff. value | std. error | $t$ | $p > |z|$ | coeff. [0.025 | coeff. 0.975] |
|---|---|---|---|---|---|---|---|---|---|
| $\beta_S$ (for $S$) | 0.1 | 0.908 | 0.764 | -0.248 | 0.008 | -30.976 | 0.000 | -0.264 | -0.233 |
| | 0.01 | 0.940 | 0.761 | -0.416 | 0.011 | -36.706 | 0.000 | -0.438 | -0.393 |
| | 0.001 | 0.931 | 0.782 | -0.553 | 0.017 | -32.507 | 0.000 | -0.586 | -0.519 |
| | 0.0001 | 0.865 | 0.628 | -0.697 | 0.031 | -22.274 | 0.000 | -0.758 | -0.635 |
| | 0.00001 | 0.862 | 0.400 | -0.802 | 0.040 | -20.311 | 0.000 | -0.880 | -0.725 |
| $\beta_C$ (for $C$) | 0.1 | 0.908 | 0.764 | 0.060 | 0.015 | 3.955 | 0.000 | 0.030 | 0.089 |
| | 0.01 | 0.940 | 0.761 | 0.169 | 0.021 | 7.941 | 0.000 | 0.127 | 0.211 |
| | 0.001 | 0.931 | 0.782 | 0.297 | 0.032 | 9.303 | 0.000 | 0.234 | 0.360 |
| | 0.0001 | 0.865 | 0.628 | 0.371 | 0.059 | 6.328 | 0.000 | 0.255 | 0.486 |
| | 0.00001 | 0.862 | 0.400 | 0.580 | 0.076 | 7.679 | 0.000 | 0.431 | 0.729 |
| $\beta_0$ (intercept) | 0.1 | 0.908 | 0.764 | 0.029 | 0.032 | 0.913 | 0.362 | -0.034 | 0.093 |
| | 0.01 | 0.940 | 0.761 | -0.118 | 0.045 | -2.613 | 0.010 | -0.208 | -0.029 |
| | 0.001 | 0.931 | 0.782 | -0.295 | 0.068 | -4.345 | 0.000 | -0.429 | -0.161 |
| | 0.0001 | 0.865 | 0.628 | -0.387 | 0.125 | -3.104 | 0.002 | -0.633 | -0.141 |
| | 0.00001 | 0.862 | 0.400 | -0.683 | 0.159 | -4.288 | 0.000 | -0.996 | -0.369 |

Figure A.5 plots the performance of the variant similar to Figure 4a in the main text.

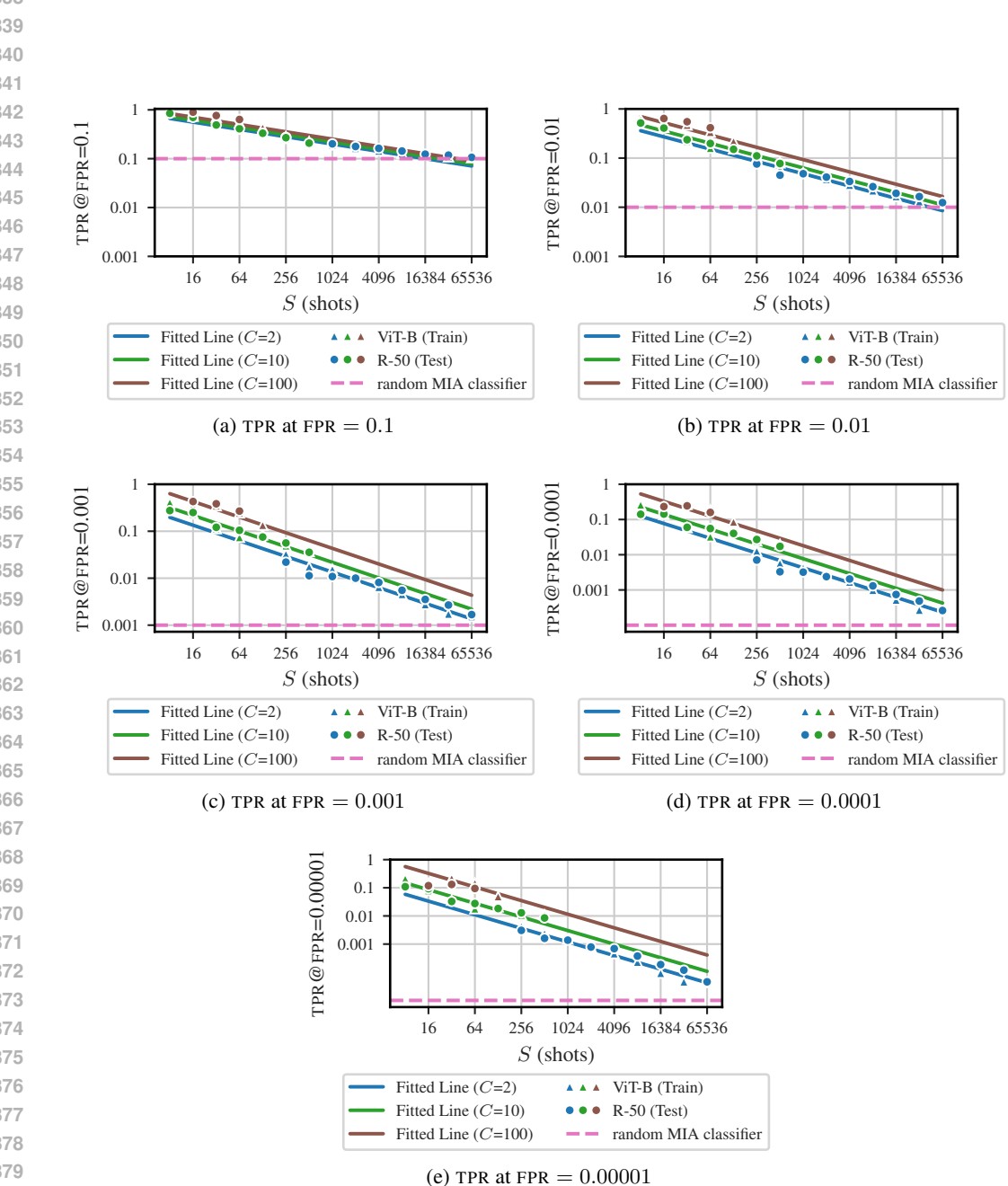

Figure A.5: Predicted MIA vulnerability as a function of $S$ (shots) using a model based on Equation (A130) fitted Table A3 (ViT-B). The triangles show the median TPR for the train set (ViT-B; Table A3) and circle the test set (R-50; Table A4) over six seeds. Note that the triangles and dots for $C = 10$ are for EuroSAT.

## F.3 EMPIRICAL RESULTS FOR RMIA

Figures A.6 to A.8 report additional results for RMIA Zarifzadeh et al. (2024).

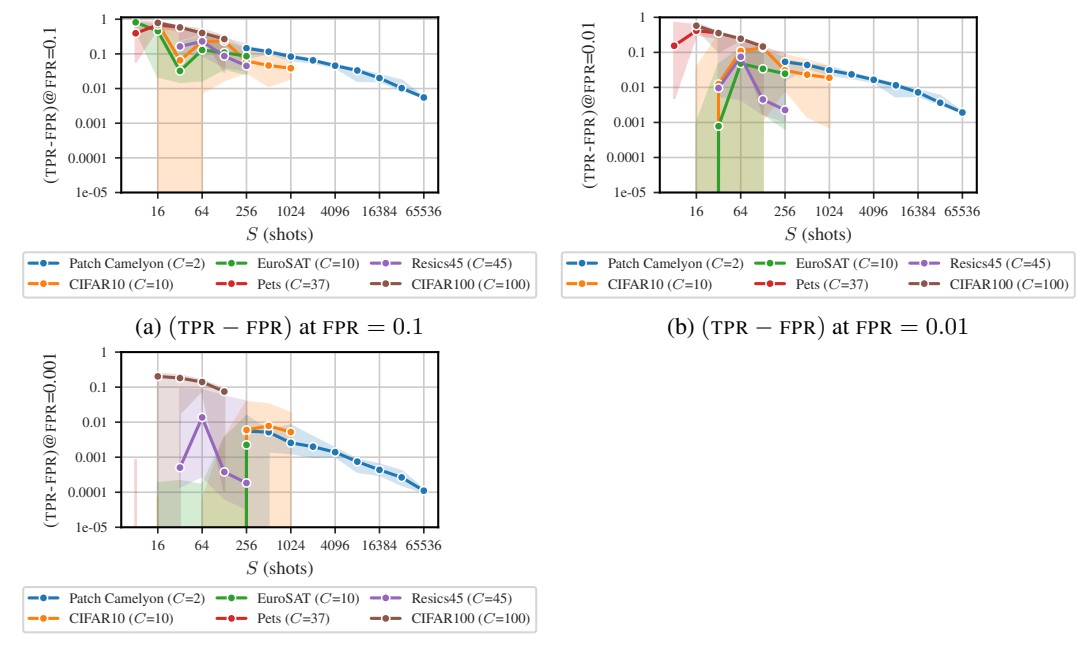

(a) $(\text{TPR} - \text{FPR})$ at $\text{FPR} = 0.1$        (b) $(\text{TPR} - \text{FPR})$ at $\text{FPR} = 0.01$

(c) $(\text{TPR} - \text{FPR})$ at $\text{FPR} = 0.001$

Figure A.6: RMIA (Zarifzadeh et al., 2024) vulnerability ($\text{TPR} - \text{FPR}$ at fixed $\text{FPR}$) as a function of $S$ (shots) when attacking a ViT-B Head fine-tuned without DP on different datasets. We observe at power-law relationship but especially at low $\text{FPR}$ the relationship is not as clear as with LiRA (compare to Figure A.2). The solid line displays the median and the error bars the minimum of the lower bounds and maximum of the upper bounds for the Clopper-Pearson CIs over six seeds.

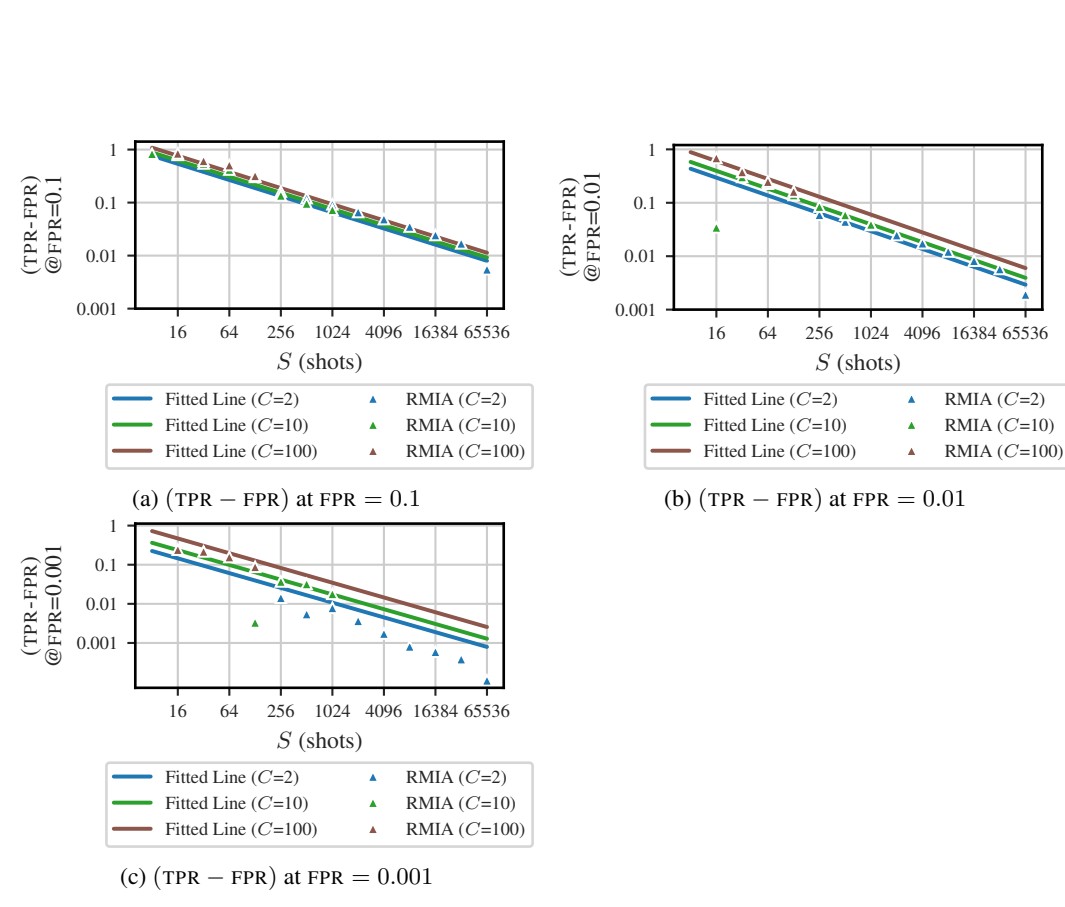

(a) $(\text{TPR} - \text{FPR})$ at $\text{FPR} = 0.1$

(b) $(\text{TPR} - \text{FPR})$ at $\text{FPR} = 0.01$

(c) $(\text{TPR} - \text{FPR})$ at $\text{FPR} = 0.001$

Figure A.7: Predicted MIA vulnerability $((\text{TPR} - \text{FPR})$ at $\text{FPR})$ based on LiRA vulnerability data as a function of $S$ (shots) in comparison to observed RMIA (Zarifzadeh et al., 2024) vulnerability on the same settings. The triangles show the highest TPR when attacking (ViT-B Head) with RMIA over six seeds (datasets: Patch Camelyon, EuroSAT and CIFAR100). Especially at $\text{FPR} = 0.1$ the relationship behaves very similar for both MIAs, but RMIA shows more noisy behavior at lower FPR.

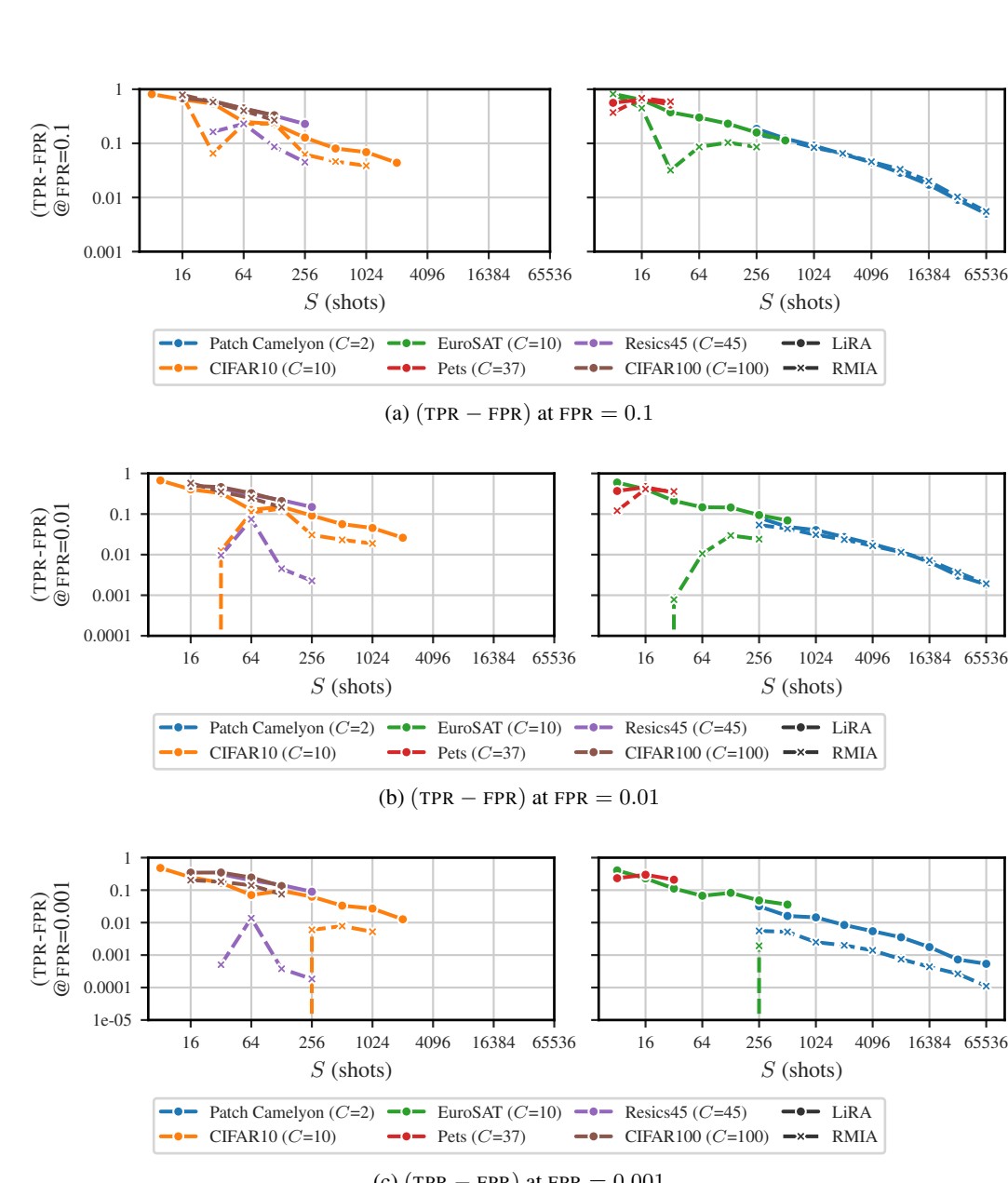

(a) $(\mathrm{TPR} - \mathrm{FPR})$ at $\mathrm{FPR} = 0.1$

(b) $(\mathrm{TPR} - \mathrm{FPR})$ at $\mathrm{FPR} = 0.01$

(c) $(\mathrm{TPR} - \mathrm{FPR})$ at $\mathrm{FPR} = 0.001$

Figure A.8: LiRA and RMIA vulnerability $((\mathrm{TPR} - \mathrm{FPR}))$ as a function of shots $(S)$ when attacking a ViT-B Head fine-tuned without DP on different datasets. For better visibility, we split the datasets into two panels. We observe the power-law for both attacks, but the RMIA is more unstable than LiRA. The lines display the median over six seeds.

