# OpenReview forum: "Impact of Dataset Properties on Membership Inference Vulnerability of Deep Transfer Learning"
_ICLR.cc/2025/Conference — Submitted to ICLR 2025_

### Official Review · Reviewer_bJQu · 2024-10-20

**Soundness:** 3
**Presentation:** 2
**Contribution:** 1
**Rating:** 3
**Confidence:** 4

**Summary:**

The study investigates privacy vulnerabilities through Membership Inference Attacks (MIA), with a particular focus on how dataset size affects model privacy. Through detailed per-sample evaluations, the researchers discovered that models become more resistant to privacy attacks as the number of examples per class increases, following a power law distribution. To further examine this phenomenon, the researchers evaluated the findings using few-shot learning and developed a regression model to predict a model's vulnerability to MIA based on dataset properties, such as the number of classes and examples per class.

**Strengths:**

- The work presents comprehensive experiments on privacy vulnerabilities using two SOTA MIA methods. It conducts a fine-grained analysis of vulnerabilities at the individual sample level.
- The findings are robustly validated across diverse architectures and datasets.
- The study provides theoretical analyses of a power-law relationship between dataset properties (e.g., shots per class) and MIA vulnerability.

**Weaknesses:**

1. The paper needs to clarify its motivation. While it discusses transfer learning and few-shot learning, it's unclear whether these techniques are studied as potential privacy protection methods or whether the authors simply examine their privacy risks. The introduction introduces these learning approaches but doesn't explicitly position them against existing differential privacy (DP) defenses. This makes it difficult to understand whether the main goal is to propose new privacy protection strategies or to analyze privacy vulnerabilities in these methods.

2. The paper's organization should be reversed. It would be clearer to show the experimental results first and then explain the theory behind them rather than the other way around.

3. The authors need to revise their claimed contributions. For example, their first contribution about analyzing privacy attacks at the individual sample level isn't new - other researchers have already done similar work (such as [1], [2]).

4. It is unsurprising that larger datasets exhibit lower overall vulnerability to MIA, as models trained on larger datasets tend to rely less on individual samples, reducing the risk of memorization.

5. The authors predicted the overall dataset vulnerability using a regression model with properties such as the number of classes and examples per class. While it offers useful insights,  individual-level vulnerabilities are often more actionable for user privacy protections. It's much more meaningful to investigate the per-sample vulnerabilities. However, considering varying dataset sizes, it may be challenging to derive consistent per-sample results, as the same sample could behave as an outlier in some datasets but not in others.

[1] Carlini, Nicholas, et al. "The privacy onion effect: Memorization is relative." Advances in Neural Information Processing Systems 35 (2022): 13263-13276.

[2] Wen, Rui, Michael Backes, and Yang Zhang. "Understanding Data Importance in Machine Learning Attacks: Does Valuable Data Pose Greater Harm?." arXiv preprint arXiv:2409.03741 (2024).

**Questions:**

see above

---

> ### Author Response · Authors · 2024-11-21
> **Response to Reviewer bJQu**
>
> We sincerely appreciate your review and useful comments. We address each comment below.
>
> > The paper needs to clarify its motivation. While it discusses transfer learning and few-shot learning, it's unclear whether these techniques are studied as potential privacy protection methods or whether the authors simply examine their privacy risks. The introduction introduces these learning approaches but doesn't explicitly position them against existing differential privacy (DP) defenses. This makes it difficult to understand whether the main goal is to propose new privacy protection strategies or to analyze privacy vulnerabilities in these methods.
>
> Our goal is to analyze MIA vulnerability in the fine-tuning of deep neural networks. Using few-show learning itself is not for privacy protection. Rather, we focus on the fine-tuning setting because it is widely used in practical applications. In addition, fine-tuning is also useful when a large dataset of labeled examples is not available, which would be the case for privacy-sensitive applications. We clarified the introduction in the revised version.
>
> > The paper's organization should be reversed. It would be clearer to show the experimental results first and then explain the theory behind them rather than the other way around.
>
> We organized the paper in this way because theoretical analysis presented in Sec. 3 entails more general results than the empirical results in Sec. 4.
>
> > The authors need to revise their claimed contributions. For example, their first contribution about analyzing privacy attacks at the individual sample level isn't new - other researchers have already done similar work (such as [1], [2]).
>
> Thank you for pointing out this. The first contribution did not mean that we focused on sample-level privacy risk, but rather we derived the closed-form LiRA/RMIA vulnerability at individual sample level. We clarified the sentence in the revised version.
>
> > It is unsurprising that larger datasets exhibit lower overall vulnerability to MIA, as models trained on larger datasets tend to rely less on individual samples, reducing the risk of memorization.
>
> As the reviewer suggests, it is indeed unsurprising that the MIA vulnerability decreases as the number of examples per class increases. However, the rate of change in this trend is not trivial. To the best of our knowledge, there is no prior work that identified the explicit rate (i.e. power-law relationship in our work). The power-law has a pragmatically useful implication that a practitioner could reduce and estimate the MIA vulnerability of a model by adding more examples in the training set.
>
> > The authors predicted the overall dataset vulnerability using a regression model with properties such as the number of classes and examples per class. While it offers useful insights, individual-level vulnerabilities are often more actionable for user privacy protections. It's much more meaningful to investigate the per-sample vulnerabilities. However, considering varying dataset sizes, it may be challenging to derive consistent per-sample results, as the same sample could behave as an outlier in some datasets but not in others.
>
> We agree that per-sample privacy risk should be a focused concern for practical applicability. In the simplified model (Sec. 3.4), the per-example MIA vulnerability also follows the power-law relationship with respect to the number of examples per class. That is, the rate of decrease in the vulnerabilities of individual examples, including outliers, would also follow the power-law.

---

> > ### Comment · Reviewer_bJQu · 2024-11-21
> >
> > Thanks for the reply. However, I would like to maintain my score as I feel the manuscript's contribution appears limited.  As mentioned above, it is not supervised that larger datasets naturally show lower vulnerability to MIAs. Furthermore, existing research already covers privacy attack analysis at the individual sample level of training data. While the authors examine the rate of decrease in individual vulnerabilities as the dataset size increases, there are additional factors that can influence the vulnerabilities, such as the presence of outliers (see citation [1] in the initial comment). Focusing solely on dataset size is too limited and impractical for real-world applications.

---

### Official Review · Reviewer_Kg6S · 2024-10-25

**Soundness:** 4
**Presentation:** 3
**Contribution:** 3
**Rating:** 8
**Confidence:** 3

**Summary:**

This paper investigates the relationship between inherent properties of a dataset, such as the number of classes and the number of samples belonging to each class, and its impact on privacy. Privacy is examined through the perspective of membership inference attacks (MIAs), which seek to determine if a sample has been used to train a model of interest. The authors present theoretical results that establish a power-law relation between dataset properties and vulnerability to MIAs, and support their claims with experimental evaluations.

****
Updated score after examining author response.
*****

**Strengths:**

1. Membership inference is an important attribute that can explain vulnerability of machine learning models to privacy attacks. The extensive theoretical results help provide generalizable and explainable interpretations of impacts of multiple characteristics of datasets on model vulnerability.

2. Theoretical results are presented in a manner that is easy to follow for an informed reader, and supported by detailed proofs in the Appendix.

3. Extensive experiments on multiple datasets validate theoretical claims. The datasets have been carefully chosen to encompass different numbers of classes and image domains.

4. The work presented has the potential for high impact beyond traditional computer science, for example, balancing the requirements of various regulatory regimes such as GDPR with a need to obtain `useful' information from possibly sensitive data.

**Weaknesses:**

1. The authors indicate that the paper considers two types of MIAs. There have been a wide variety of MIAs examined in the literature, including some which may not rely on confidence scores (e.g., label-based MIAs). Is there a claim (perhaps speculative) that the authors can make about similar relationships for other classes of MIAs?

2. The experiments focus on datasets and models related to image classification. Can the results be extended to other domains, such as audio or video? Multi-modality can strengthen the scope of the proposed contribution.

3. The authors point out that analyses and experimental results rely on an assumption that the datasets are balanced. Some perspective on what might happen if this is not the case can shed more light on real-world applicability of the work.

4. A more detailed discussion or comparison of the effect of dataset characteristics on other aspects of privacy- e.g., differential privacy, which has an extensive literature and supported by theoretical guarantees- will strengthen the submission. Could the authors comment on this?

**Questions:**

Please see Weaknesses above.

---

> ### Author Response · Authors · 2024-11-21
> **Response to Reviewer Kg6S**
>
> Thank you for your time and effort to review our paper and provide insightful feedback. We address each comment below.
>
> > The authors indicate that the paper considers two types of MIAs. There have been a wide variety of MIAs examined in the literature, including some which may not rely on confidence scores (e.g., label-based MIAs). Is there a claim (perhaps speculative) that the authors can make about similar relationships for other classes of MIAs?
>
> As the reviewer suggests, we only focus on score-based MIAs, while there are other threat models. We are not claiming that the observed power-law relationship would hold for white-box attacks. However, we emphasize that score-based MIAs are a realistic strong threat model. In addition, for other weaker attacks, including label-only MIAs, it is sufficient to have the power-law for the currently known strongest MIAs (LiRA and RMIA) which provides the upper bound for the vulnerability to other weaker MIAs.
>
> > The experiments focus on datasets and models related to image classification. Can the results be extended to other domains, such as audio or video? Multi-modality can strengthen the scope of the proposed contribution.
>
> Image classification has been widely studied in privacy research. We believe that limiting the experiments to this simple setting would sufficiently support our theoretical findings. However, as the reviewer suggests, it would be important to expand the scope of application domains in future research.
>
> > The authors point out that analyses and experimental results rely on an assumption that the datasets are balanced. Some perspective on what might happen if this is not the case can shed more light on real-world applicability of the work.
>
> If the power-law relationship we identified in this work holds in a real-world application, for unbalanced datasets, a practitioner could reduce and estimate an upper-bound for the MIA vulnerability of the entire dataset by increasing the number of examples in the smallest class. Although theoretically the MIA vulnerability should not depend on the number of classes, our empirical results suggest that there could be some dependence. Therefore, when the number of classes is large, a practitioner would need to add more examples than in the case of a small number of classes to achieve the same privacy protection. We added this discussion in Sec. 5 of the revised version.
>
> > A more detailed discussion or comparison of the effect of dataset characteristics on other aspects of privacy- e.g., differential privacy, which has an extensive literature and supported by theoretical guarantees- will strengthen the submission. Could the authors comment on this?
>
> Prior works studied how datasets with different properties (e.g. the number of examples per class) are affected by applying DP, for example. We have added references to the related work paragraph in Sec. 1. However, these prior works on DP susceptibility do not consider the rate of change in the susceptibility as the dataset properties change.  On the other hand, our focus is on the explicit relationship between the privacy risk and dataset properties (i.e. power-law) in the non-DP setting. To the best of our knowledge, there is no prior work that explicitly established the effect of dataset properties such as the number of examples per class on MIA vulnerability or DP susceptibility.

---

### Official Review · Reviewer_j6WZ · 2024-11-01

**Soundness:** 3
**Presentation:** 4
**Contribution:** 1
**Rating:** 3
**Confidence:** 4

**Summary:**

The paper proposes a theoretical result that links the success of membership inference attacks (MIAs) to the number of training data points per class. This theoretical link is shown for the RMIA and LiRA attack, and empirical evaluation is performed through fine-tuning vision encoders on multiple downstream datasets.

**Strengths:**

First, I'd like to note that I did enjoy reading the paper: it is clearly written, nicely presented, and well executed. Particularly the plots are extremely visually appealing.

Overall, the attempts to provide some interesting theoretical underpinning for a long known and well studied problem.

**Weaknesses:**

The core weaknesses of the paper can be grouped as follows:

**Studying a well established problem, while not taking prior work into account**

The underlying causes of MIA risks have been studied in a long line of work since 2017. The findings made in the paper, showing that vulnerability decreases as you have more samples per class, and with fewer classes have been made already. Hence, the expected impact and relevance of the presented findings is negligible. Given the enormous amounts of prior work, it is particularly surprising that the paper does not even have a related work section.

This section could start at early work. E.g. [1] (2021) has an extremely related approach where they assign privacy risk scores that implicitly expresses a similar phenomenon: with the probability of a sample being part of the training set yielding a per point privacy risk: here, it is over classes. Additionally, they add a very valuable angle (not considered in this present work here excessively): Classes with less members have on average lower probability: "the class labels with high generalization errors tend to have higher privacy risk scores".
Given the results by [2], this seems extremely crucial to analyze. For the given work, I am wondering how much this power law connection really holds when data is not well curated. We can imagine a really "malformed" class that consists of extremely rare and outlier data points with high variance. How much will having more of those really help reduce membership risk?
Starting from these works, a related work section could help put the given work into context.

Additionally, another line of work that needs to be taken into account is the one around [3,4,5] where it has been shown that with less data points per class, the class is more vulnerable and that privacy attacks have disparate success.

**Generality of Results**

While LiRA and RMIA are *current* state-of-the-art, it is, undoubtably reasonable to derive the bounds for those concrete attacks. However, the results are presented "as is", i.e., without embedding them into any broader context and without generalization. This makes it questionable, how well the results will age, i.e., how much can we benefit from them in 2-3 years, when LiRA and RMIA are obsolete and have been replaced by new and more powerful attacks.

In a similar vein, a suggestion would be to improve the writing in Section 3.4. The results are interesting, but it is not discussed what they mean: "Hence we can have the LiRA vulnerability arbitrarily close to zero in this non-DP setting by simply increasing S" -> this seems to mean that if every class has more and more train data, the membership risk decreases. Formulating such intermediate "checkpoints" in plain English would be genuinely helpful to position the results.

**Evaluation done for fine-tuning might skew the results**

Overall, I am not convinced that the chosen evaluation setup is sensible for the presented results. The paper considers transfer learning where the heavy lifting is done by a pre-trained vision encoder. However, the pre-training data itself has a distribution that will play a role for the vulnerability of the fine-tuning data. Data points that have high overlap with the classes / concepts present in the pre-training will have a significantly different vulnerability than data from an entirely different distribution. Hence, to underline the theoretical results, the correct approach, would have been training from scratch.

While the paper tries to argue that the pre-training encoder has no influence ("Figure 4a shows that the regression model is robust to a change of the feature extractor"), this statement is too broad and not backed up by results: The experiments were conducted on only two encoders, both pre-trained on ImageNet. These encoders will have the same biases, and will expose the fine-tuning data equally. If any, to make such a statement, it would be required to assess more encoders, especially ones trained on entirely different distributions, including non-object centric dataset.

The fact, that the expectations do not match the from-scratch training ("underestimates the vulnerability of the from-scratch trained target models") is another indicator that comparing with fine-tuning is not the right approach.


**Minor Comments**

- In the intro, the paper says: "and apply two state-of-the-art MIAs", it would be good to cite them there already for the readers to understand what to expect
- Would it make sense to add the values from the theoretical model in Figure 3, such that one can compare how much the empirical one deviates?

**References**

[1] Song, L., & Mittal, P. (2021). Systematic evaluation of privacy risks of machine learning models. In 30th USENIX Security Symposium (USENIX Security 21) (pp. 2615-2632).

[2] Yeom, Samuel, Irene Giacomelli, Matt Fredrikson, and Somesh Jha. "Privacy risk in machine learning: Analyzing the connection to overfitting." In 2018 IEEE 31st computer security foundations symposium (CSF), pp. 268-282. IEEE, 2018.

[3] Suriyakumar, Vinith M., Nicolas Papernot, Anna Goldenberg, and Marzyeh Ghassemi. "Chasing your long tails: Differentially private prediction in health care settings." In Proceedings of the 2021 ACM Conference on Fairness, Accountability, and Transparency, pp. 723-734. 2021.

[4] Bagdasaryan, Eugene, Omid Poursaeed, and Vitaly Shmatikov. "Differential privacy has disparate impact on model accuracy." Advances in neural information processing systems 32 (2019).

[5] Kulynych, Bogdan, Mohammad Yaghini, M. Cherubin, G. Veale, and C. Troncoso. "Disparate vulnerability: On the unfairness of privacy attacks against machine learning." In 22st Privacy Enhancing Technologies Symposium. 2022.

**Questions:**

NA

---

> ### Author Response · Authors · 2024-11-21
> **Response to Reviewer j6WZ 1/2**
>
> We sincerely appreciate your review and very constructive feedback. We address each point below.
>
> > Studying a well established problem, while not taking prior work into account
>
> Thank you for pointing out relevant papers. Song & Mittal (2021) empirically demonstrate the positive correlation between the privacy risk and the generalization error of each class. However, Song & Mittal (2021) define the privacy risk of a target example as the posterior probability of the target being in the training set given the attacker's observation.Thus, their findings are not directly comparable to our MIA vulnerability metric TPR at a low FPR. Yeom et al. (2018) theoretically and empirically show similar results. Nonetheless, Yeom et al. (2018) evaluates MIA performance by maximizing TPR - FPR, not for a low FPR. Therefore, this is also not comparable to our findings.  In addition,  both of these two papers are not concerned with dataset properties that our work studies, fixing the dataset size and the number of classes. Although the number of examples per class explored in our work would indeed be related to the model's generalization, Yeom et al. (2018) in fact demonstrate that generalization error is not necessary for MIA success. We have added these references to the related work paragraph in Sec. 1.
>
> Suriyakumar et al. (2021) empirically show that classes with less examples have less influence on prediction performance when DP is applied. Bagdasaryan et al. (2019) demonstrate similar results that smaller subgroups are more strongly affected by DP in terms of prediction accuracy. Kulynych et al. (2022) show that there are greater disparities in MIA vulnerability among subgroups of training data when the subgroup size is smaller. Although these works suggest that the number of data points in a class is indeed related to the privacy susceptibility of that class, they do not mention the rate of changes as the number of examples in a class changes. Moreover, since they do not consider the privacy vulnerability at a low FPR, the findings are not directly comparable to our work. We have added these references to the related work paragraph in the revised version.
>
>
> > the results are presented "as is", i.e., without embedding them into any broader context and without generalization. This makes it questionable, how well the results will age, i.e., how much can we benefit from them in 2-3 years, when LiRA and RMIA are obsolete and have been replaced by new and more powerful attacks.
>
> We emphasize that score-based attacks are a strong realistic threat model, and LiRA with infinitely many shadow models is the optimal attack against the simplified model (Sec 3.4) by the Neyman-Pearson lemma. We clarified this in the revised version. Nevertheless, we acknowledge the limitation that there could be stronger attacks in the future for which the vulnerability behaves differently. We revised the limitation paragraph in Sec. 5.
>
> > “Hence we can have the LiRA vulnerability arbitrarily close to zero in this non-DP setting by simply increasing S” -> this seems to mean that if every class has more and more train data, the membership risk decreases. Formulating such intermediate “checkpoints” in plain English would be genuinely helpful to position the results.
>
> The expression is revised. Thank you for your suggestion.
>
> > Evaluation done for fine-tuning might skew the results
>
> We focused on fine-tuning of neural networks because it is widely used in practical applications. Fine-tuning is also important when a large number of labeled examples are not available in the application domain, which would often be the case for privacy-sensitive applications. Moreover, from-scratch training has been extensively studied, while research on the privacy risk of fine-tuned models remains relatively scarce. That said, we acknowledge that models trained from scratch are likely to be more vulnerable, but this is out of scope in our work. Additionally, our simplified model (Sec. 3.4.) is designed to resemble the linear (HEAD) classifier often used for fine-tuning.

---

> > ### Author Response · Authors · 2024-11-21
> > **Response to Reviewer j6WZ 2/2**
> >
> > > the pre-training data itself has a distribution that will play a role for the vulnerability of the fine-tuning data. Data points that have high overlap with the classes / concepts present in the pre-training will have a significantly different vulnerability than data from an entirely different distribution.
> >
> > Thanks for the question. Generally, the fact that fine-tuning using the Head works indicates that the pre-training data contains images that lead to useful features also for the fine-tuning but we consider datasets where the concepts/classes not overlapping between pre-training [1] and fine-tuning (Patch Camelyon, EuroSAT, Resics45) and where prior work [2] has noted that the difficulty of transferring is not lower (CIFAR-100, Resics45). So at least Resisc45 is not close to the pre-training data based on both measures.
> >
> > When looking at the concepts/classes of the pre-training dataset ImageNet-21k [1] the classes/concepts are overlapping with certain fine-tuning datasets, e.g., CIFAR-10, CIFAR-100 or Pets, but we were unable to confirm this for Patch Camelyon (histopathologic scans of lymph node sections) or remote sensing datasets (EuroSAT, Resics45).
> >
> > There are not really any good measures for understanding the overlap between pre-training and fine-tuning data as noted by prior work [2]. The same prior work [2] suggests using an empirical measure to determine the "transfer-difficulty" between pre-training and fine-tuning. In their Table 1 they report that the transfer-difficulty is low for CIFAR-10, EuroSAT, Patch Camelyon, Pets and medium for CIFAR-100, Resics45.
> >
> > [1] ImageNet-21k  class names: https://storage.googleapis.com/bit_models/imagenet21k_wordnet_lemmas.txt
> >
> > [2] Tobaben, M., Shysheya, A., Bronskill, J. F., Paverd, A., Tople, S., Zanella-Beguelin, S., ... & Honkela, A. On the Efficacy of Differentially Private Few-shot Image Classification. TMLR 2023.
> >
> >
> > > In the intro, the paper says: "and apply two state-of-the-art MIAs", it would be good to cite them there already for the readers to understand what to expect
> >
> > We added the citations. Thank you for your suggestion.
> >
> > > Would it make sense to add the values from the theoretical model in Figure 3, such that one can compare how much the empirical one deviates?
> >
> > Thank you for your suggestion. We have modified Figure 3 so that it has theoretical values from the simplified model.

---

> > > ### Comment · Reviewer_j6WZ · 2024-11-24
> > > **Thank you to the authors for their response**
> > >
> > > I thank the authors for their response and implementing the feedback. It addresses everything that could be addressed to improve the paper on a low level. The high-level concerns, especially the narrow scope of the work remain, hence I will keep my score.

---

### Official Review · Reviewer_A23d · 2024-11-03

**Soundness:** 3
**Presentation:** 3
**Contribution:** 2
**Rating:** 3
**Confidence:** 4

**Summary:**

This paper investigates the impact of dataset properties on membership inference attacks (MIAs). The authors analyze two state-of-the-art MIA methods, LiRA and RMIA, by deriving a relationship between dataset properties (e.g., number of examples per class) and MIA vulnerability. They introduce a simplified model for membership inference and demonstrate that a power-law relationship exists between per-example MIA vulnerability and the dataset's properties. The authors also fit a regression model to predict MIA vulnerability, providing insights into privacy risks for fine-tuned classifiers.

**Strengths:**

1.	The paper formalizes the effect of dataset properties on membership inference vulnerability for two advanced MIA methods (LiRA and RMIA), enriching the analytical framework for MIA.
2.	The study includes a comprehensive and well-designed set of experiments across multiple datasets.

**Weaknesses:**

1.	In the abstract and introduction, the authors emphasize that their focus is on the privacy analysis of fine-tuned image classification models. However, sections 2 and 3 lack specific analyses that are tailored to fine-tuned neural networks, relying instead on a general theoretical model.
2.	The authors need to clarify the differences in membership inference vulnerability between standard image classification models and fine-tuned models. This would strengthen the argument for fine-tuning as a key focus in their work.
3.	As the regression model is presented as one of the main contributions, the methods section would benefit from a more detailed description of this model, its design, and its implementation process.
4.	Although LiRA and RMIA are currently the best methods, it is also necessary to analyze other MIA methods to validate the generalizability of the proposed arguments.

**Questions:**

1.	As the authors mention in the introduction, prior work has already explored the impact of the number of examples per class (shots) on MIA vulnerability. How does this study differ from previous work, and what new insights does it provide beyond existing findings?
2.	While LiRA and RMIA represent state-of-the-art approaches in MIA, it is also essential to assess the generalizability of the presented results to other MIA techniques. How might other methods differ in vulnerability patterns under the same dataset conditions?
3.	As a widely used method for privacy protection, it is necessary to discuss the impact of differential privacy on the arguments presented in this paper.

---

> ### Author Response · Authors · 2024-11-21
> **Response to Reviewer A23d**
>
> Thank you for your time and effort to review our manuscript and provide insightful feedback. We address the comments below.
>
> > The authors need to clarify the differences in membership inference vulnerability between standard image classification models and fine-tuned models
>
> Fine-tuning is increasingly used in practical applications. It is also important when a large dataset of labeled examples is not available in the application domain, which would often be the case for privacy-sensitive applications. Moreover, while from-scratch training has been extensively studied, research on the privacy risk of fine-tuned models remains relatively scarce. Therefore, we focus on the fine-tuned models, leaving from-scratch training out of scope. We now clarify this motivation in the revised version.
>
> > As the authors mention in the introduction, prior work has already explored the impact of the number of examples per class (shots) on MIA vulnerability. How does this study differ from previous work, and what new insights does it provide beyond existing findings?
>
> This work differs from prior work in that we identify the **rate** of the impact, namely, the power-law relationship. The literature suggests that, if we have more data in the training set, then the model is less vulnerable to MIA. However, to the best of our knowledge prior works did not explicitly identify how much the MIA vulnerability decreases, as the size of the training dataset grows. With the evidence of the power-law relationship between MIA vulnerability and the number of examples per class, machine learning practitioners or analysts would be able to estimate the membership privacy risk and how large a training set would be needed to mitigate the risk. We clarified this in the revised introduction.
>
> > While LiRA and RMIA represent state-of-the-art approaches in MIA, it is also essential to assess the generalizability of the presented results to other MIA techniques. How might other methods differ in vulnerability patterns under the same dataset conditions?
>
> In this work we focused on LiRA and RMIA that are state-of-the-art score-based MIAs. For white-box attacks, the vulnerability might behave differently. However, score-based attacks are a more realistic threat model, thus being our focus in this work. For other existing weaker attacks, including label-only attacks, it is sufficient to have the power-law relationship for the currently known strongest attacks as an upper bound. For potential future MIA, we acknowledge the limitation that there might be stronger attacks in the future such that the vulnerability behaves differently. That said, it should be noted that LiRA with infinitely many shadow models is the optimal attack against the simplified model (Sec. 3.4) by the Neyman-Pearson lemma. Thus, as long as the simplified model reflects real-world models, the power-law is expected to be observed. We now clearly state the optimality of the simplified model in the introduction as well as in Sec. 3.4.
>
> > As a widely used method for privacy protection, it is necessary to discuss the impact of differential privacy on the arguments presented in this paper.
>
> Our work investigates the MIA vulnerability (TPR at a law FPR) in non-DP settings. Applying DP gives an upper bound on MIA vulnerability, and thus we would not need to worry about dataset properties. However, this comes at a cost. Therefore, it is important to understand when DP is necessary. We stress this point in the introduction.

---

### Meta-Review · Area_Chair_6MBF · 2024-12-18

**Metareview:**

Summary: This paper studies how the number of examples per class and the number of classes impact the membership inference attacks (MIA), using two state-of-the-art MIA attacks, LiRA and RMIA. The paper finds a power-law relationship between those properties and MIA vulnerability. Empirical evaluation is performed through fine-tuning vision encoders on multiple downstream datasets.

Strengths:
1. What matters in MIA vulnerability is an important topic to the ICLR community. The paper identifies the number of examples per class and the number of classes as the key factors, with theoretical underpinning.
2. The paper is well-written and organized, with rich experimental results.

Weaknesses:
Reviewers have common concerns on the paper:
1. Though LiRA and RMIA are two state-of-the-art MIA methods, there are more attacks in the literature. The paper does not test on other MIA methods to validate the generalizability of the proposed arguments.
2. The paper highlights that it focuses on fine-tuned models. Reviewers have concerns on it as reviewers believe the training data of the pre-trained models might also impact MIA vulnerability. Instead, the authors are encouraged to do the test on models trained from scratch.
3. Discussion of related work is missing in the paper. Reviewers raised concern as they find previous work has sent a similar message as this paper (e.g., the number of examples per class and the number of classes matter in MIA vulnerability). Therefore, reviewers believe the paper fails to provide enough new messages to the community.

Three out of four reviewers vote for strong rejection with a score of 3. Following the majority vote, AC would recommend rejection. AC would encourage the authors to take the above weaknesses into account in the next revised version.

**Additional Comments On Reviewer Discussion:**

It appears that reviewers have different opinions on the paper: Reviewer Kg6S is positive about the paper with a score of 8, while the other reviewers are extremely negative about the paper with a score of 3. The authors have written a rebuttal, but it seems that the rebuttal fails to change negative reviewers' mind and they would like to maintain their scores. AC also checks the strengths in Reviewer Kg6S's review, and find most of them to be minor. Following the majority vote, AC would recommend rejection.

---

### Decision · Program_Chairs · 2025-01-22

Reject